# TOWARDS SELF-ROBUST LLMs: INTRINSIC PROMPT NOISE RESISTANCE VIA CoIPO

**Xin Yang**[1,3]*, **Letian Li**[2], **Abudukelimu Wuerkaixi**[2,3]*, **Xuxin Cheng**[3],
**Cao Liu**[3], **Ke Zeng**[3], **Xunliang Cai**[3†], **Wenyuan Jiang**[4†]

[1]School of Mathematical Sciences, Zhejiang University, Hangzhou, China
[2]Tsinghua University, China
[3]Meituan LongCat Interaction Team, China
[4]D-INFK, ETH Zürich, Switzerland

## ABSTRACT

Large language models (LLMs) have demonstrated remarkable and steadily improving performance across a wide range of tasks. However, LLM performance may be highly sensitive to prompt variations especially in scenarios with limited openness or strict output formatting requirements, indicating insufficient robustness. In real-world applications, user prompts provided to LLMs often contain imperfections, which may undermine the quality of the model's responses. To address this issue, previous work has primarily focused on preprocessing prompts, employing external tools or even LLMs to refine prompt formulations in advance. However, these approaches overlook the intrinsic robustness of LLMs, and their reliance on external components introduces additional computational overhead and uncertainty. In this work, we propose a **Co**ntrastive Learning-based **I**nverse Direct **P**reference **O**ptimization (CoIPO) method that minimizes the discrepancy between the label-aligned logits produced by the model under a clean prompt and its noisy counterpart, and conduct a detailed analysis using mutual information theory. We augment the FLAN dataset by constructing paired prompts, each consisting of a clean prompt and its corresponding noisy version for training. Additionally, to evaluate the effectiveness, we develop NoisyPromptBench, a benchmark enhanced and derived from the existing PromptBench. Experimental results conducted on NoisyPromptBench demonstrate that our proposed method achieves a significant improvement in average accuracy over the current state-of-the-art approaches. The source code of CoIPO, pair-wise FLAN datasets, and NoisyPromptBench have already been released on https://github.com/vegetable-yx/CoIPO.

## 1 INTRODUCTION

In recent years, the advancement of LLM has achieved exceptional and progressively better performance in various natural language processing tasks, including text generation (Devlin et al., 2019), machine translation (Ouyang et al., 2022), and logical reasoning (Cheng et al., 2025), due to their strong contextual understanding and reasoning abilities. However, their practical utility is limited by the high sensitivity to input prompts. In settings with restricted openness (e.g., mathematical problem solving (Pei et al., 2025; Tang et al., 2025),

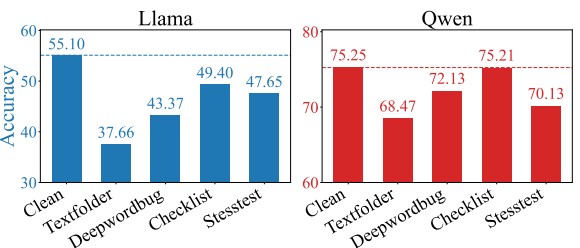

Figure 1: The performance degradation of the models under different perturbation scenarios. "Llama" denotes the Llama2-7B model and "Qwen" represents the Qwen2.5-7B model.

code generation (Jiang et al., 2024)) or strict output formats (e.g., XML, JSON), even small prompt

---

*This work was done during an internship at Meituan.
†Xunliang Cai and Wenyuan Jiang are corresponding authors. Correspondence to: wenyjiang@ethz.ch

variations, such as spelling mistakes, word substitutions, or stylistic changes, can markedly degrade output quality. This highlights a core weakness in the robustness of LLMs under prompt perturbations (Fig. 1), decreased by as much as 17.44% and 6.78% in TextFolder, respectively.

In real-world applications, user prompts rarely meet the "perfection" required for optimal model performance. Spelling mistakes (e.g., typing "clasify" instead of "classify"), semantic deviations (e.g., confusing "diagnosis" with "investigation" in medical tasks), or irrelevant additions (e.g., appending "Interesting fact: cats sleep for most of their lives." to a math problem (Rajeev et al., 2025)) can all degrade LLM responses. Such imperfections limit the reliability of LLMs in practical scenarios such as customer service dialogues and intelligent office assistants.

To mitigate this issue, existing research has focused on prompt preprocessing and repair: external tools (e.g., grammar checkers (Fan et al., 2023), terminology-normalization systems (Fan et al., 2024)) or LLM-based rewriting techniques (Zhou et al., 2022; Xu et al., 2024; Pryzant et al., 2023) are used to detect and correct noisy prompts effectively.

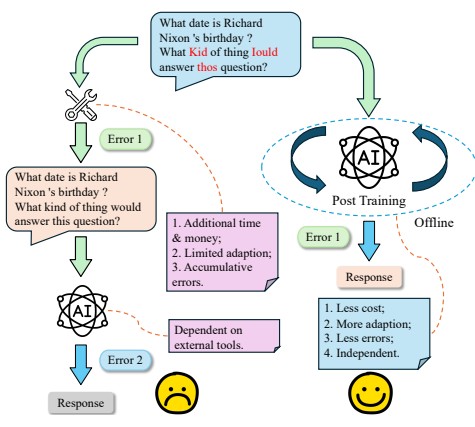

However, these approaches face three key limitations as shown in Fig 2. First, reliance on external pre-processing tools inevitably incurs additional computational overhead, financial cost, and deployment complexity, making them less practical in real-world applications. Second, these methods suffer from pipeline cascading errors, such as multiple intermediate stages that amplify inaccuracies, causing the final output to drift away from the original user intent. Third, they overlook the intrinsic robustness within the model itself, leaving it dependent on auxiliary components rather than fully exploiting its own capacity to handle perturbed or imperfect inputs. Moreover, most existing noise evaluation datasets (e.g., PromptBench (Zhu et al., 2023)) only support single-

Figure 2: External tools incur additional time and monetary costs, limit adaptability, and introduce cumulative errors. In contrast, self-robustness requires only offline training, without the associated challenges.

step perturbations, substantially limiting their effectiveness in simulating real-world scenarios.

To address the above issues, we propose Contrastive Learning based Inverse Direct Preference Optimization (CoIPO), a method designed to enhance the robustness of LLMs against prompt perturbations from within the model itself, which is shown in Fig 3. The core idea is to construct paired samples of clean prompts and noisy prompts, and during training, minimize the alignment discrepancy in the model's output for semantically equivalent prompts, while maximizing the alignment discrepancy for semantically different prompts. To adapt to our method, first, we construct a paired FLAN dataset by generating noisy prompts for each clean prompt in the original dataset through character-level, word-level, or sentence-level perturbations. This yields paired clean–noisy samples that serve as the foundation for contrastive learning in CoIPO. Second, we developed the NoisyPromptBench benchmark based on PromptBench by enhancing four categories of perturbations: DeepWordBug, TextFolder, CheckList, and StrssTest, thus providing a standardized evaluation framework for prompt robustness. Finally, we use mutual information to justify the effectiveness of our method, and we conduct comparative experiments on NoisyPromptBench against the current state-of-the-art intrinsic robustness method, CoIN. Results show that CoIPO achieves an average accuracy improvement of 3.64% across all noise types, with the highest gain of 4.18% under the TextFolder perturbation scenario, demonstrating its strong advantage in enhancing intrinsic model robustness under situations where various types of noises are in prompts.

The main contributions of this paper can be summarized as follows:

1. **The CoIPO Framework:** We introduce *CoIPO*, a novel method to enhance prompt robustness by post-training, thereby eliminating the need for external preprocessing components.

2. **The Paired FLAN Dataset and NoisyPromptBench:** We construct the *Paired FLAN* dataset and develop the *NoisyPromptBench* benchmark, providing high-quality training data and standardized evaluation tools for robustness research under noisy prompts.

3. **Empirical validation and theoretical analysis:** We provide extensive experimental validation of CoIPO's effectiveness across diverse noise scenarios, complemented by an information-theoretic analysis that offers a theoretical justification for its robustness and a new pathway for enhancing LLM reliability.

## 2 RELATED WORK

**Prompt automatic optimization** has become a core research focus for enhancing the adaptability and performance of LLMs across diverse tasks, with methods primarily divided into gradient-based and gradient-free paradigms. Gradient-based methods are limited to open-source LLMs with accessible weights, as they depend on gradient information or additional parameter training. One major line of work is soft prompt tuning, where Li & Liang (2021), Lester et al. (2021), and Wang et al. (2023b) avoid modifying native weights by applying gradient descent to learn task-specific "soft prompt" parameters. Another direction is gradient-based discrete prompt search: Shin et al. (2020) and Wen et al. (2023) exploit model gradients to identify effective discrete prompts. Besides, reinforcement learning has been incorporated into gradient-based methods: RLPrompt (Deng et al., 2022) introduces a rewriting mechanism, while Zhang et al. (2023) treats task performance as a reward signal, enabling iterative optimization of discrete prompts.

Gradient-free methods address the lack of gradient access in closed-source LLMs, relying instead on sampling, rewriting, heuristic algorithms, and LLM-driven self-optimization. For iterative sampling and diverse generation, Zhou et al. (2022) introduced an "iterative sampling–selection" framework augmented with Monte Carlo search, Xu et al. (2024) applied Gibbs sampling, and Pryzant et al. (2023) adopted beam search, all enhancing prompt diversity. Rewriting-based approaches include Xu et al. (2022), which used back-translation for semantically similar prompts, Prasad et al. (2023), which optimized expressions through phrase deletion and replacement, and Guo et al. (2024), Fernando et al. (2024), which applied genetic algorithms to evolve prompt populations. A major trend is LLM-driven self-optimization: Zhou et al. (2022), Pryzant et al. (2023), and Yang et al. (2023) leveraged LLMs to rewrite prompts based on natural language feedback; SHUM et al. (2023) and Zhang et al. (2022) optimized prompts by generating chain-of-thoughts; and Zhou et al. (2022) (APE) further enabled end-to-end automatic prompt generation directly from input–output pairs. Nevertheless, all these methods assume clean prompts and largely overlook the impact of noisy input.

**Noisy prompt preprocessing** has gained increasing attention in recent studies, as it addresses the vulnerability of LLMs to noisy or perturbed prompts, driving research on techniques to mitigate performance degradation. Several studies have demonstrated LLM fragility under prompt noise: Gu et al. (2023), Wang et al. (2024) applied perturbations at character, word, and sentence levels; Zhu et al. (2023) conducted large-scale adversarial attacks across multiple dimensions; Sun et al. (2023) showed that paraphrased instructions degrade performance; and Liang et al. (2023) found inconsistent instruction placement also reduces effectiveness.

To mitigate these issues, preprocessing-based solutions have been proposed. BAT (Shi et al., 2024) adopts adversarial training to generate perturbation-resistant prompts, RoP (Mu et al., 2025) leverages noisy–clean prompt pairs for denoising, and PromptAgent (Wang et al., 2023a) applies Monte Carlo Tree Search with LLM feedback to iteratively refine prompts. Despite their effectiveness, these methods suffer from inherent limitations: they rely on additional preprocessing components, adding inference cost and latency. More critically, these approaches overlook the possibility of enhancing LLMs' intrinsic robustness, leaving open the question of how to enable models to directly process noisy prompts without relying on external modules.

## 3 METHODOLOGY

### 3.1 BACKGROUND

To formulate the problem of noisy prompt scenarios, we denote suboptimal prompts as $P'_x$ and optimal prompts as $\tilde{P}$. In reality, $\tilde{P}$ is nearly impossible—or at least extremely difficult—to find. Instead, we aim to design a series of prompts $(\hat{P}_0, \hat{P}_1, \ldots)$ such that $\hat{P}_x$ approximates $\tilde{P}$ as closely

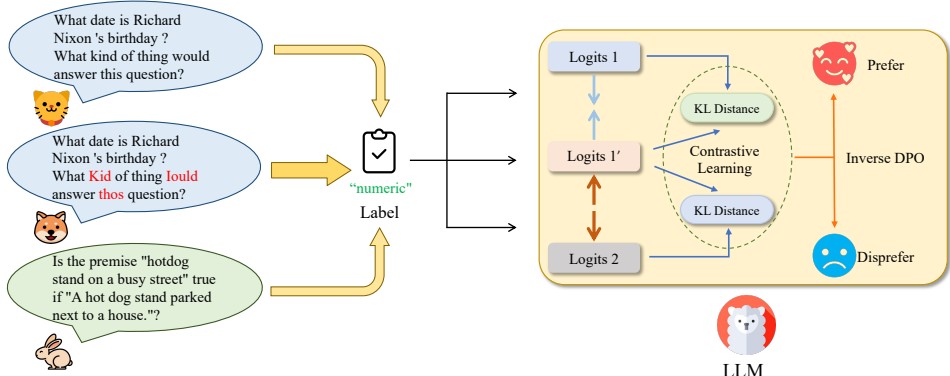

Figure 3: Framework of CoIPO: The clean prompt and its corresponding perturbed version (in blue text), along with an unrelated prompt (in green text), are first concatenated with the label. The logits are then computed by the LLM for each. Logits 1 is preferred, while Logits 2 is dispreferred. Subsequently, based on the principles of contrastive learning, the KL divergence similarity between Logits 1 and Logits 2 relative to Logits 1' is calculated, with the goal of maximizing the similarity to the former and minimizing the similarity to the latter.

as possible. For a given threshold $r$, we consider a prompt $\hat{P}_x$ to be a clean prompt if

$$\left| F(\tilde{P}, \hat{P}_x) \right| \leq r, \tag{1}$$

meaning it enables the LLM to achieve satisfactory performance. Conversely, a suboptimal prompt $P'_x$ satisfies

$$\left| F(\tilde{P}, P'_x) \right| > r. \tag{2}$$

Here, $F$ measures the performance gap induced by different prompts on a specific model for a specific task, and $r$ can be defined as the *perturbation radius*, which measures the degree of perturbation. Fig. 4 intuitively illustrates different perturbation radii of different perturbations. We view $P'_x$ as a perturbed variant of $\hat{P}_x$, expressed as

$$P'_x = \hat{P}_x + N, \tag{3}$$

where $N$ represents a perturbation operation that alters $\hat{P}_x$ in various ways—both in type and degree—degrading its quality. The existence of such perturbations $N$ leads to a wide range of $P'_x$ instances, which in turn degrade user experience and reduce problem-solving success rates.

Inspired by works such as PromptRobust (Zhu et al., 2023), we select several common perturbation operations as representative cases: DeepWordBug (Gao et al., 2018), TextFooler (Jin et al., 2020), CheckList (Ribeiro et al., 2020), and StressTest (Naik et al., 2018). The first two represent character-level and word-level perturbations, respectively, while the latter two involve sentence-level transformations. Specific examples are provided in Appendix C. These common types of noise can degrade model performance to varying degrees, as shown in Fig. 1.

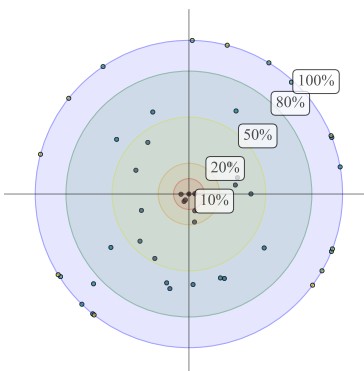

Figure 4: Schematic diagram of perturbation radii. Dots represent various perturbations, and their distance from the center of the circle is the perturbation radius. The perturbation radius is quantified by the performance degradation rate.

### 3.2 CoIPO

Building on the above background, we propose CoIPO (Fig. 3) that integrates preference learning and contrastive learning. In DPO (Rafailov et al., 2023), the standard approach is to compare the log-probability differences of alternative outputs given the same input $x$, and then optimize the model according to preference signals. In our setting, however, the goal is to fix the ground-truth label $y$ and compare the conditional probabilities of different prompts with respect to this label. In other words, rather than comparing "different outputs under

the same input," we compare "different inputs under the same output." This motivates the notion of Inverse DPO (invDPO). Formally, we first abstractly define a comparison function $D$:

$$\mathcal{L}_{\text{invDPO}} = -D\Big(\hat{P}_2 \Big| P_1', y_1\Big) + D\Big(\hat{P}_1 \Big| P_1', y_1\Big), \tag{4}$$

where $\hat{P}_1$ and $\hat{P}_2$ denote two clean prompts for different tasks, $P_1'$ represents the noisy version of $\hat{P}_1$, and $y_1$ is the corresponding ground-truth label sequence for $\hat{P}_1$ and $P_1'$. D means the chosen probability when given $P_1'$ and $y$.

To implement $D$, we adopt a contrastive learning perspective: the similarity of the logits distributions on the label tokens serves as a proxy for probability. Let $\theta$ denote the model parameters, and define the function $g_\theta(\cdot)$ as the model's forward mapping from an input token sequence to the sequence of logits over the vocabulary for each time step. Specifically, for an input sequence $S = P \oplus y$ (concatenating prompt $P$ and label $y$), the output logits are

$$\ell_{1:T}(S) = g_\theta(S) \in \mathbb{R}^{T \times |\mathcal{V}|}, \tag{5}$$

where $T$ is the sequence length and $\mathcal{V}$ is the vocabulary. A masking operator $\mathcal{M}_y(\cdot)$ is applied to retain only the logits corresponding to the label part $y$, discarding the prompt positions. The per-token conditional distribution is then

$$p_t^{(P,y)} = \text{softmax}\Big(\mathcal{M}_y(\ell_t(P \oplus y))\Big), \quad t \in \mathcal{T}_y(S) \tag{6}$$

where $\mathcal{T}_y(S)$ denotes the index set of label tokens. Accordingly, $D$ is instantiated as the sequence-level KL divergence:

$$\begin{aligned}
D(P \mid P_{\text{ref}}, y) &= \sum_{t \in \mathcal{T}_y} \text{KL}\Big( \text{softmax}(\mathcal{M}_y(\ell_t(P_{\text{ref}} \oplus y))) \parallel \text{softmax}(\mathcal{M}_y(\ell_t(P \oplus y))) \Big) \\
&= \sum_{t \in \mathcal{T}_y} \text{KL}\Big( p_t^{(P_{\text{ref}},y)} \parallel p_t^{(P,y)} \Big).
\end{aligned} \tag{7}$$

Finally, the overall loss function is given by:

$$\mathcal{L} = -\sum_{t \in \mathcal{T}_{y_1}} \text{KL}\big(p_t^{(P_1',y_1)} \| p_t^{(\hat{P}_2,y_1)}\big) + \sum_{t \in \mathcal{T}_{y_1}} \text{KL}\big(p_t^{(P_1',y_1)} \| p_t^{(\hat{P}_1,y_1)}\big), \tag{8}$$

where the first item measures the divergence between the noisy prompt $P_1'$ and the clean prompt $\hat{P}_2$ in terms of their label logits, while the second measures the divergence between $P_1'$ and $\hat{P}_1$. By minimizing $\mathcal{L}$, the model is encouraged to reduce the gap between $\hat{P}_1$ and its noisy counterpart $P_1'$, while simultaneously enlarging the gap with $\hat{P}_2$. This contrastive formulation effectively enforces robustness against noisy prompts and improves prompt quality alignment.

### 3.3 FROM MUTUAL INFORMATION TO COIPO

We further interpret CoIPO by analyzing how the method enhances the information content that distinguishes correct task-prompt pairs from incorrect ones under noisy conditions.

The mutual information (Belghazi et al., 2018) $I(X; Y)$ measures the amount of information that one random variable contains about another, while the entropy (Sepúlveda-Fontaine & Amigó, 2024) $H(X)$ quantifies the uncertainty or information content of a random variable. The conditional mutual information $I(X; Y \mid Z)$ captures the mutual dependence between $X$ and $Y$ given knowledge of $Z$, and the conditional entropy $H(X \mid Y)$ represents the remaining uncertainty in $X$ after observing $Y$. Let $Y$ denote the ground-truth label sequence for a specific task, and consider two prompts: $\hat{P}_1$ (the clean prompt for one task) and $\hat{P}_2$ (a clean prompt for a different task). The noisy prompt $P_1' = \hat{P}_1 + N$ serves as our reference point for comparison.

The key insight is that CoIPO aims to maximize the *Discriminative Information* that the correct clean prompt $\hat{P}_1$ provides about the label $Y$, relative to an incorrect prompt $\hat{P}_2$, when both are evaluated

against the noisy reference $P_1'$. Formally, we define the *Relative Mutual Information gain* as:

$$\Delta I = I(Y; \hat{P}_1 \mid P_1') - I(Y; \hat{P}_2 \mid P_1'), \tag{9}$$

where $I(Y; \hat{P} \mid P_1')$ represents the mutual information between the label $Y$ and prompt $\hat{P}$, conditioned on the noisy reference $P_1'$. This conditioning captures the idea that we evaluate both clean prompts in the context of the noisy prompt. Using the definition of conditional mutual information:

$$I(Y; \hat{P} \mid P_1') = H(Y \mid P_1') - H(Y \mid \hat{P}, P_1'), \tag{10}$$

we obtain:

$$
\begin{aligned}
\Delta I &= [H(Y \mid P_1') - H(Y \mid \hat{P}_1, P_1')] - [H(Y \mid P_1') - H(Y \mid \hat{P}_2, P_1')] \\
&= H(Y \mid \hat{P}_2, P_1') - H(Y \mid \hat{P}_1, P_1').
\end{aligned} \tag{11}
$$

Since the true conditional distributions are unknown, we approximate them using the model's predictions. Specifically, we use the model's output distribution under the noisy prompt as a reference: $q(y) = p_\theta(y \mid P_1')$. This choice is natural because $P_1'$ represents the actual noisy input the model encounters during training. The empirical conditional entropy under this reference distribution becomes:

$$\tilde{H}_q(Y \mid \hat{P}) = -\mathbb{E}_{y \sim q} \log p_\theta(y \mid \hat{P}), \tag{12}$$

where we drop the explicit conditioning on $P_1'$ in the notation for simplicity, understanding that the reference distribution $q$ implicitly encodes this conditioning. Therefore, the empirical mutual information difference is:

$$\Delta \tilde{I}_q = \tilde{H}_q(Y \mid \hat{P}_2) - \tilde{H}_q(Y \mid \hat{P}_1) = \mathbb{E}_{y \sim q}\Big[ \log p_\theta(y \mid \hat{P}_1) - \log p_\theta(y \mid \hat{P}_2) \Big]. \tag{13}$$

By the definition of KL divergence, this can be rewritten as:

$$\Delta \tilde{I}_q = \mathrm{KL}\big(q \| p_\theta(\cdot \mid \hat{P}_2)\big) - \mathrm{KL}\big(q \| p_\theta(\cdot \mid \hat{P}_1)\big). \tag{14}$$

By substituting our per-token distributions $p_t^{(P,y)}$ from Eq. 6, and noting that $q(y)$ corresponds to the token-wise distributions $\{p_t^{(P_1',y)}\}_{t \in \mathcal{T}_y}$, we obtain:

$$\Delta \tilde{I}_q = \sum_{t \in \mathcal{T}_{y_1}} \mathrm{KL}\big(p_t^{(P_1',y_1)} \| p_t^{(\hat{P}_2,y_1)}\big) - \sum_{t \in \mathcal{T}_{y_1}} \mathrm{KL}\big(p_t^{(P_1',y_1)} \| p_t^{(\hat{P}_1,y_1)}\big). \tag{15}$$

Maximizing this relative mutual information gain is equivalent to minimizing its negation. Comparing with the CoIPO loss function in Eq. 8, we see that

$$\mathcal{L}_{\mathrm{CoIPO}} = -\Delta \tilde{I}_q. \tag{16}$$

This demonstrates that minimizing the CoIPO loss is equivalent to maximizing the relative mutual information gain, providing a principled information-theoretic foundation for our approach. The method effectively learns to extract more discriminative information from the correct prompt while reducing the information shared with incorrect prompts, all evaluated in the context of noisy observations.

## 4 EXPERIMENTS

**Settings.** We conducted experiments on the widely used FLAN dataset (Wei et al., 2022), which is a large-scale dataset encompassing a diverse range of natural language processing (NLP) tasks such as natural language inference, commonsense reasoning, sentiment analysis, and paraphrase identification. In our work, we selected 25 subsets of this dataset that feature deterministic answers; for each entry within these subsets, we first generated a clean prompt by randomly choosing one from a set of pre-defined, well-formed instruction templates, and then applied perturbations to this clean prompt to create a corresponding noise prompt, thereby forming a clean-noise prompt pair. To

enhance the generalization capability of the trained model, the type of perturbation used to generate each noise prompt was selected randomly.

For the model architecture, we employed Alpaca (Taori et al., 2023)—an instruction-tuned model built on LLaMA-7B (Touvron et al., 2023) and trained on the 52k Self-Instruct dataset, and Qwen2.5-7B; all models were trained on the aforementioned augmented FLAN dataset (i.e., the dataset with clean-noise prompt pairs) using a learning rate of $1 \times 10^{-4}$, a batch size of 64, and a maximum sequence length of 256, with all experiments performed on NVIDIA A100 GPUs.

Table 1: Performance Comparison of Llama Under Different Perturbations and Datasets. Acc means accuracy score (%), Diff means score difference compared to clean (%).

| Perturbation | Method | MNLI | | MRPC | | QNLI | | QQP | | SST2 | | Avg | |
|---|---|---|---|---|---|---|---|---|---|---|---|---|---|
| | | Acc | Diff | Acc | Diff | Acc | Diff | Acc | Diff | Acc | Diff | Acc | Diff |
| Clean | Base | 51.00 | / | 62.04 | / | 52.54 | / | 29.87 | / | 80.04 | / | 55.10 | / |
| | SFT | 56.13 | / | 41.04 | / | 46.54 | / | 58.62 | / | 84.04 | / | 57.28 | / |
| | COIN | 61.58 | / | 56.92 | / | 53.79 | / | 50.67 | / | 86.38 | / | 61.87 | / |
| | CoIPO | 61.50 | / | 64.50 | / | 56.21 | / | 66.04 | / | 86.75 | / | **67.00** | / |
| TextFolder | Base | 39.96 | 11.04 | 42.12 | 19.92 | 41.00 | 11.54 | 13.17 | 16.71 | 52.04 | 28.00 | 37.66 | 17.44 |
| | SFT | 54.83 | 1.29 | 39.79 | 1.25 | 43.08 | 3.46 | 51.33 | 7.29 | 80.29 | 3.75 | 53.87 | **3.41** |
| | COIN | 52.83 | 8.75 | 51.71 | 5.21 | 48.75 | 5.04 | 42.12 | 8.54 | 85.13 | 1.25 | 56.11 | 5.76 |
| | CoIPO | 58.00 | 3.50 | 64.38 | 0.13 | 51.12 | 5.08 | 52.17 | 13.88 | 85.42 | 1.33 | **62.22** | 4.78 |
| DeepWordBug | Base | 42.62 | 8.38 | 47.29 | 14.75 | 48.00 | 4.54 | 17.50 | 12.37 | 61.46 | 18.58 | 43.37 | 11.73 |
| | SFT | 52.83 | 3.29 | 39.58 | 1.46 | 44.37 | 2.17 | 51.04 | 7.58 | 82.71 | 1.33 | 54.11 | **3.17** |
| | COIN | 51.92 | 9.67 | 51.34 | 5.58 | 45.13 | 8.67 | 39.46 | 11.21 | 86.29 | 0.08 | 54.83 | 7.04 |
| | CoIPO | 57.33 | 4.17 | 61.50 | 3.00 | 51.04 | 5.17 | 48.54 | 17.50 | 86.54 | 0.21 | **60.99** | 6.01 |
| CheckList | Base | 47.71 | 3.29 | 55.62 | 6.42 | 47.67 | 4.88 | 20.25 | 9.62 | 75.75 | 4.29 | 49.40 | 5.70 |
| | SFT | 54.13 | 2.00 | 40.25 | 0.79 | 40.33 | 6.21 | 57.17 | 1.46 | 81.46 | 2.58 | 54.67 | 2.61 |
| | COIN | 59.58 | 2.00 | 55.08 | 1.83 | 52.00 | 1.79 | 50.21 | 0.46 | 85.04 | 1.33 | 60.38 | **1.48** |
| | CoIPO | 59.38 | 2.12 | 63.67 | 0.83 | 53.58 | 2.62 | 64.50 | 1.54 | 85.83 | 0.92 | **65.39** | 1.61 |
| StressTest | Base | 45.46 | 5.54 | 51.33 | 10.71 | 53.33 | -0.79 | 21.00 | 8.87 | 67.12 | 12.92 | 47.65 | 7.45 |
| | SFT | 50.17 | 5.96 | 39.54 | 1.50 | 50.12 | -3.58 | 50.04 | 8.58 | 78.50 | 5.54 | 53.68 | 3.60 |
| | COIN | 54.46 | 7.12 | 58.33 | -1.42 | 55.58 | -1.79 | 45.75 | 4.92 | 84.96 | 1.42 | 59.82 | **2.05** |
| | CoIPO | 55.71 | 5.79 | 66.29 | -1.79 | 57.08 | -0.88 | 55.04 | 11.00 | 85.33 | 1.42 | **63.89** | 3.11 |
| Avg | Base | 45.35 | 7.06 | 51.68 | 12.95 | 48.51 | 5.04 | 20.36 | 11.89 | 67.28 | 15.95 | 46.64 | 10.58 |
| | SFT | 53.62 | **3.14** | 40.04 | 1.25 | 44.89 | **2.06** | 53.64 | **6.23** | 81.40 | 3.30 | 54.72 | **3.20** |
| | COIN | 56.08 | 6.89 | 54.68 | 2.80 | 51.05 | 3.43 | 45.64 | 6.28 | 85.56 | 1.02 | 58.60 | 4.08 |
| | CoIPO | **58.38** | 3.89 | **64.07** | **0.54** | **53.81** | 3.00 | **57.26** | 10.98 | **85.98** | **0.97** | **63.90** | 3.88 |

**Benchmark.** For the benchmark construction, we have implemented enhancements based on PromptBench. Specifically, PromptBench encompasses a diverse array of datasets and seven distinct types of perturbations. From these, we selected five representative datasets and four perturbation types, which are designed to cover the character-level, word-level, and sentence-level granularities, respectively. Each dataset is accompanied by eight predefined clean prompt templates. These templates can be directly concatenated with the corresponding dataset samples for immediate utilization in experiments. To better simulate the inherent randomness of perturbation intensity in real-world scenarios, we perform random sampling multiple times for each perturbation type (see Appendix E for detailed sampling rules). This design enables a more comprehensive and rigorous evaluation, thereby ensuring the credibility of our experimental conclusions.

**Baselines.** A series of comprehensive experiments was carried out on the Llama and Qwen model families. For each model, we first assessed the performance of its base version to establish a baseline. We then evaluated three distinct training strategies: direct fine-tuning, which incorporates noise prompts but omits any comparative or reference-based learning and instead performs supervised fine-tuning (SFT) directly using labels; the COIN-based (Yan et al., 2024) approach; and our proposed CoIPO method. Through systematic experimentation and cross-comparison of these methods across the two model families, the effectiveness of our CoIPO method is empirically validated.

## 4.1 EVALUATION OF CoIPO

As presented in Table 1 and Table 2, our method demonstrates consistent superiority across nearly all datasets and perturbation settings: it not only attains the highest accuracy but also exhibits a relatively smaller accuracy drop when prompts are perturbed.

For Llama, CoIPO consistently achieves the highest accuracy across all datasets, surpassing COIN by 5.3%, SFT by 9.18%, and Base by 17.26% on average. When exposed to perturbed prompts, CoIPO exhibits a performance drop of only 3.88%, which is slightly higher than SFT's 3.20%. However,

given that SFT attains a much lower accuracy under clean prompts (54.72%), its overall robustness remains limited. In contrast, CoIPO not only achieves the best accuracy under every perturbation type but also maintains relatively small degradation compared to its clean performance.

For Qwen, CoIPO consistently outperforms the other three methods across almost all datasets. Specifically, its average accuracy surpasses COIN by 1.97%, SFT by 6.6%, and Base by 11.21%. Under perturbed prompts, CoIPO exhibits the smallest performance degradation, with only a 0.54% drop, ranking the lowest among all methods. Across different types of perturbations, CoIPO not only achieves the highest accuracy but also maintains one of the smallest relative drops compared to clean prompts. Taken together, these results demonstrate that CoIPO consistently delivers state-of-the-art performance across diverse datasets and perturbations, achieving both superior accuracy and strong robustness. Additionally, we have considered other metrics.

Table 2: Performance comparison of Qwen under different perturbations and Datasets.

| Perturbation | Method | MNLI | | MRPC | | QNLI | | QQP | | SST2 | | Avg | |
|---|---|---|---|---|---|---|---|---|---|---|---|---|---|
| | | Acc | Diff | Acc | Diff | Acc | Diff | Acc | Diff | Acc | Diff | Acc | Diff |
| Clean | Base | 83.29 | / | 61.16 | / | 71.92 | / | 70.67 | / | 89.21 | / | 75.25 | / |
| | SFT | 80.21 | / | 74.12 | / | 71.96 | / | 72.54 | / | 90.87 | / | 77.94 | / |
| | COIN | 87.71 | / | 76.46 | / | 77.08 | / | 80.83 | / | 92.54 | / | 82.93 | / |
| | CoIPO | 86.75 | / | 78.83 | / | 78.75 | / | 84.08 | / | 91.00 | / | **83.88** | / |
| TextFolder | Base | 76.79 | 6.50 | 58.33 | 2.83 | 67.62 | 4.29 | 61.50 | 9.17 | 78.08 | 11.12 | 68.47 | 6.78 |
| | SFT | 80.83 | -0.62 | 71.08 | 3.04 | 67.92 | 4.04 | 68.46 | 4.08 | 86.67 | 4.21 | 74.99 | 2.95 |
| | COIN | 87.58 | 0.12 | 75.88 | 0.58 | 73.33 | 3.75 | 71.79 | 9.04 | 91.54 | 1.00 | 80.03 | 2.90 |
| | CoIPO | 87.21 | -0.46 | 72.62 | 6.21 | 77.46 | 1.29 | 84.04 | 0.04 | 90.00 | 1.00 | **82.27** | **1.62** |
| DeepWordBug | Base | 77.17 | 6.13 | 53.75 | 7.42 | 70.88 | 1.04 | 70.83 | -0.17 | 88.04 | 1.17 | 72.13 | 3.12 |
| | SFT | 79.83 | 0.37 | 72.42 | 1.71 | 76.79 | -4.83 | 71.29 | 1.25 | 89.83 | 1.04 | 78.03 | **-0.09** |
| | COIN | 87.08 | 0.62 | 76.83 | -0.38 | 79.62 | -2.54 | 73.87 | 6.96 | 91.33 | 1.21 | 81.75 | 1.18 |
| | CoIPO | 86.38 | 0.37 | 78.83 | 0.00 | 81.33 | -2.58 | 82.08 | 2.00 | 90.96 | 0.04 | **83.92** | -0.03 |
| CheckList | Base | 81.58 | 1.71 | 64.83 | -3.67 | 65.17 | 6.75 | 74.21 | -3.54 | 90.25 | -1.04 | 75.21 | 0.04 |
| | SFT | 80.62 | -0.42 | 73.71 | 0.42 | 71.50 | 0.46 | 66.54 | 6.00 | 91.42 | -0.55 | 76.76 | 1.18 |
| | COIN | 88.08 | -0.38 | 75.54 | 0.92 | 74.38 | 2.71 | 78.21 | 2.62 | 92.67 | -0.12 | 81.77 | 1.15 |
| | CoIPO | 86.04 | 0.71 | 79.38 | -0.54 | 78.79 | -0.04 | 84.29 | -0.21 | 91.37 | -0.37 | **83.97** | **-0.09** |
| StressTest | Base | 82.75 | 0.54 | 52.17 | 9.00 | 69.12 | 2.79 | 64.12 | 6.54 | 82.50 | 6.71 | 70.13 | 5.12 |
| | SFT | 80.21 | 0.00 | 73.08 | 1.04 | 71.71 | 0.25 | 71.25 | 1.29 | 86.42 | 4.46 | 76.53 | 1.41 |
| | COIN | 86.96 | 0.75 | 76.17 | 0.29 | 74.79 | 2.29 | 74.71 | 6.12 | 91.92 | 0.62 | 80.91 | 2.02 |
| | CoIPO | 86.21 | 0.54 | 78.12 | 0.71 | 77.54 | 1.21 | 83.54 | 0.54 | 90.62 | 0.38 | **83.21** | **0.68** |
| Avg | Base | 80.32 | 3.72 | 58.05 | 3.89 | 68.94 | 3.72 | 68.27 | 3.00 | 85.62 | 4.49 | 72.24 | 3.76 |
| | SFT | 80.34 | **-0.17** | 72.88 | 1.55 | 71.97 | -0.02 | 70.02 | 3.16 | 89.04 | 2.29 | 76.85 | 1.36 |
| | COIN | **87.48** | 0.28 | 76.18 | **0.35** | 75.84 | 1.55 | 75.88 | 6.19 | **92.00** | 0.68 | 81.48 | 1.81 |
| | CoIPO | 86.52 | 0.29 | **77.56** | 1.59 | **78.77** | **-0.03** | **83.61** | **0.59** | 90.79 | **0.26** | **83.45** | **0.54** |

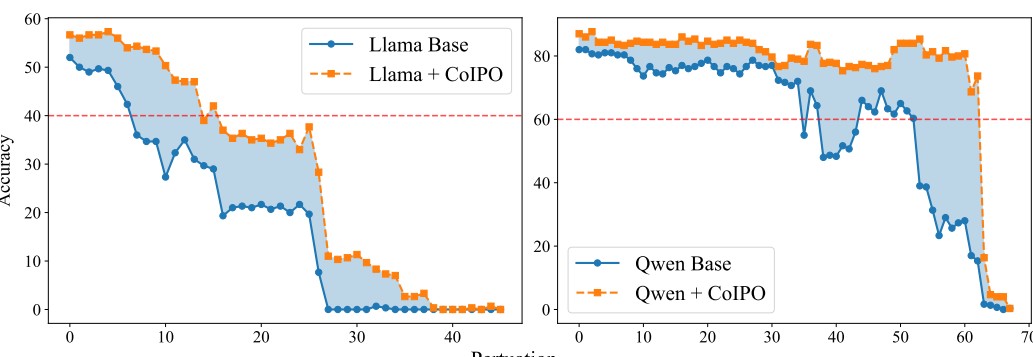

Figure 5: Trend chart illustrating the decline in performance with increasing perturbations.

**Performance under perturbation.** Let $r := d(\hat{P}, P')$ denote the perturbation radius as 3.1, i.e., the number of character edits applied to the clean prompt $\hat{P}$. We model task performance (accuracy) under perturbation as a non-increasing function of $r$:

$$\mathrm{Acc}(r) = \mathrm{Acc}(0) - \Delta(r), \tag{17}$$

where $\Delta(r)$ is the performance drop induced by perturbations. This captures the empirical observation that larger perturbation radii cause more severe degradation.

**Decoding radius.** We further define the *decoding radius* $R(a)$ as the largest perturbation radius under which performance remains at least $a$:

$$R(a) = \sup r. \quad \mathrm{Acc}(r) \geq a. \tag{18}$$

Intuitively, $R(a)$ characterizes the robustness margin: a larger $R(a)$ means the model can withstand more edits before accuracy drops to $a$.

**Results.** Figure 5 illustrates the relationship between the perturbation radii of CoIPO and Base on the MNLI dataset. As shown by the red dashed line in the figure, for a given accuracy $a$, the maximum perturbation radius $R(a)$ corresponding to CoIPO is significantly larger compared to Base. This observation highlights that CoIPO has a larger decoding radius and demonstrates superior robustness, maintaining high performance even under perturbed conditions. Such a result underscores the effectiveness of CoIPO in handling perturbations, which is a crucial aspect for improving model robustness in real-world scenarios.

## 4.2 ABLATION STUDY

We conducted detailed ablation studies to validate the effectiveness of our approach. Specifically, since our method integrates both inverse DPO and contrastive learning, we compared CoIPO against three variants: a baseline without any additional techniques (SFT), a model using only contrastive learning (CL), and a model using only inverse DPO (InvDPO), all evaluated on the same datasets.

As shown in Table 3, our method consistently achieves the highest accuracy and exhibits relatively low performance degradation on both Llama and Qwen. Specifically, CoIPO achieves accuracies of 63.90% and 83.45% on Llama and Qwen, respectively. Under perturbed prompts, Llama demonstrates a second-best performance degradation rate of 3.20%, while Qwen achieves the lowest degradation rate of 0.54%, significantly outperforming other methods. This clearly highlights the superiority of our approach. Although methods using only contrastive learning or only InvDPO do not surpass CoIPO, they still outperform the baseline SFT method, further demonstrating the effectiveness and necessity of each component in our method. More details about our ablation experiment can be found in Section F.

Table 3: Ablation experiment result. CoIPO outperforms all other methods, demonstrating the effectiveness of the approach and the necessity of each of its components.

| Model | Method | Clean | | TextFolder | | DeepWordBug | | CheckList | | StressTest | | Avg | |
|---|---|---|---|---|---|---|---|---|---|---|---|---|---|
| | | Acc | Diff | Acc | Diff | Acc | Diff | Acc | Diff | Acc | Diff | Acc | Diff |
| Llama | SFT | 57.28 | / | 53.87 | **3.41** | 54.11 | **3.17** | 54.67 | 2.61 | 53.68 | 3.60 | 54.72 | **3.20** |
| | CL | 61.87 | / | 56.11 | 5.76 | 54.83 | 7.04 | 60.38 | 1.48 | 59.82 | **2.05** | 58.60 | 4.08 |
| | InvDPO | 65.89 | / | 59.84 | 6.05 | **61.14** | 4.75 | 64.89 | **1.00** | 61.82 | 4.07 | 62.72 | 3.97 |
| | CoIPO | 67.00 | / | **62.22** | 4.78 | 60.99 | 6.01 | **65.39** | 1.61 | **63.89** | 3.11 | **63.90** | 3.88 |
| Qwen | SFT | 77.94 | / | 74.99 | 2.95 | 78.03 | -0.09 | 76.76 | 1.18 | 76.53 | 1.41 | 76.85 | 1.36 |
| | CL | 82.93 | / | 80.03 | 2.90 | 81.75 | 1.18 | 81.77 | 1.15 | 80.91 | 2.02 | 81.48 | 1.81 |
| | InvDPO | 83.31 | / | **82.41** | **0.90** | 83.50 | **-0.18** | 82.71 | 0.60 | 82.36 | 0.95 | 82.86 | 0.56 |
| | CoIPO | **83.88** | / | 82.27 | 1.62 | **83.92** | -0.03 | **83.97** | **-0.09** | **83.21** | **0.68** | **83.45** | **0.54** |

## 4.3 MODEL SCALING EXPERIMENTS

We further conduct a model size scaling study to examine whether the effectiveness of our method persists across different model capacities. In addition to Qwen2.5-7B, we evaluate 14B and 72B variants trained with the same procedure. Detailed results are provided in Table 4. As shown, our approach consistently outperforms competing methods across all model scales, aligning with the conclusions drawn above. Moreover, our method—as well as the baselines—generally follows the expected scaling trend that larger models yield better performance. These findings collectively demonstrate the robustness, superiority, and broad applicability of our approach.

## 4.4 EVALUATION ON BROADER CAPABILITY BENCHMARKS

Although our method has been extensively validated across diverse datasets, perturbation types, model families, and even model scales—consistently outperforming existing approaches—a natural and important concern remains: How does our method generalize to task types that are not included in training, such as mathematical reasoning, code generation, and open-ended generative tasks? Clearly, a robustness improvement that compromises performance on these tasks would be unacceptable.

To address this, we evaluate our method on three representative benchmarks: mathematical reasoning (GSM8K (Cobbe et al., 2021)), open-ended generation (TruthfulQA (Lin et al., 2022)), and code

Table 4: Model size scaling results for Qwen2.5 models and baselines.

| Model | Method | Clean | TextFolder | DeepWordBug | CheckList | StressTest | Avg |
|-------|--------|-------|-----------|-------------|-----------|------------|-----|
| **Qwen2.5-7B** | Base | 75.25 | 68.47 | 72.13 | 75.21 | 70.13 | 72.24 |
| | SFT | 77.94 | 74.99 | 78.03 | 76.76 | 76.53 | 76.85 |
| | COIN | 82.93 | 80.03 | 81.75 | 81.77 | 80.91 | 81.48 |
| | CoIPO | **83.88** | **82.27** | **83.92** | **83.97** | **83.21** | **83.45** |
| **Qwen2.5-14B** | Base | 80.00 | 76.30 | 74.63 | 79.42 | 76.91 | 77.45 |
| | SFT | 80.18 | 80.20 | 79.63 | 80.27 | 78.82 | 79.82 |
| | COIN | 74.87 | 69.67 | 66.43 | 75.28 | 73.09 | 71.86 |
| | CoIPO | **84.98** | **83.04** | **83.88** | **83.86** | **83.91** | **83.93** |
| **Qwen2.5-72B** | Base | 84.20 | 82.64 | 71.93 | 82.48 | 80.85 | 80.42 |
| | SFT | 84.20 | 84.42 | 84.18 | 84.10 | 83.18 | 84.02 |
| | COIN | 82.78 | 80.92 | 81.97 | 81.90 | 81.03 | 81.72 |
| | CoIPO | **85.44** | **85.47** | **84.53** | **84.68** | **84.85** | **85.00** |

generation (MBPP (Austin et al., 2021)) with lm-evaluation-harness framework (Gao et al., 2024). We compare the results of our method with the original baseline, as summarized in Table 5. We observe that, despite not being trained on these datasets, our method does not degrade performance on these tasks; in fact, the overall performance is slightly improved compared to the baseline. These results demonstrate the practicality and generalizability of our approach.

Table 5: Performance of CoIPO versus the baseline on GSM8K, TruthfulQA, and MBPP. Reported metrics include exact match, BLEU/ROUGE (n-gram and LCS overlap), and Pass@1 for code execution correctness.

| Model | Method | GSM8K Exact Match | TruthfulQA | | | | MBPP Pass@1 |
|-------|--------|-------------------|------------|------------|------------|------------|-------------|
| | | | Bleu Acc | Rouge1 Acc | Rouge2 Acc | RougeL Acc | |
| **Llama-7B** | Base | **7.20** | **34.14** | 32.43 | 25.82 | 30.96 | **18.0** |
| | CoIPO | 6.29 | 34.02 | **32.92** | **28.02** | **31.57** | 17.8 |
| **Qwen2.5-7B** | Base | 70.05 | 46.76 | **51.41** | **42.23** | **46.63** | **35.6** |
| | CoIPO | **74.37** | **47.61** | 48.59 | 41.13 | 46.51 | 32.8 |
| **Qwen2.5-14B** | Base | 75.66 | **53.37** | **54.35** | **44.68** | **51.41** | 36.4 |
| | CoIPO | **84.76** | 51.41 | 53.98 | 43.57 | 51.16 | **43.2** |
| **Qwen2.5-72B** | Base | 80.97 | **50.18** | 51.29 | **45.41** | **49.69** | 74.2 |
| | CoIPO | **86.28** | **50.18** | **51.65** | 44.31 | 48.59 | **77.0** |

## 5 DISCUSSION AND CONCLUSION

This work addresses the fundamental challenge of prompt robustness in large language models, where minimal input perturbations can cause significant performance degradation. In contrast to existing methodologies that predominantly rely on external preprocessing pipelines or post-hoc prompt repair mechanisms, our proposed CoIPO framework enhances robustness at the model level through the integration of contrastive learning principles with inverse DPO. Drawing from information-theoretic foundations, we derive CoIPO's core mechanism from a mutual information maximization perspective, which systematically minimizes logit discrepancies under adversarial noise conditions. To facilitate comprehensive training and evaluation, we construct an augmented version of the FLAN dataset incorporating contrastive prompt pairs and introduce NoisyPromptBench, a benchmark encompassing diverse perturbation taxonomies representative of real-world noise distributions.

Our empirical evaluation demonstrates that CoIPO consistently achieves superior performance relative to baseline approaches, with significant improvements in accuracy across heterogeneous noise scenarios. Furthermore, we provide granular analysis of performance degradation patterns as a function of perturbation intensity, complemented by comprehensive ablation studies isolating individual component contributions. In addition, evaluations on other benchmark tasks demonstrate that our method does not compromise the model's capabilities beyond prompt robustness; the model-size scaling experiments confirm the generality and broad applicability of our approach across models of different capacities. This research establishes a novel paradigm for intrinsic robustness enhancement in LLMs and provides theoretical and empirical foundations for developing noise-resilient foundation models suitable for deployment in noisy, real-world environments.

## ACKNOWLEDGMENT

This work is sponsored by Beijing Nova Program.

## ETHICS STATEMENT

This work improves the robustness of large language models to prompt perturbations using only publicly available datasets (e.g., FLAN, PromptBench) with automatically generated noisy variants. No human subjects, private, or sensitive data are involved. We considered potential dual-use risks, but the main contribution is to enhance reliability under natural prompt imperfections, consistent with the ICLR Code of Ethics. All code, datasets, and benchmarks are released under open licenses to encourage responsible community use.

## REPRODUCIBILITY STATEMENT

We describe all model settings, datasets, perturbation procedures, and hyperparameters in Section 4 and Appendix A. Our method is fully specified in Section 3 with theoretical justification in Section 3.3. To ensure reproducibility, we provide source code, paired FLAN datasets, and the NoisyPromptBench benchmark at https://github.com/vegetable-yx/CoIPO, enabling verification and extension of our results.

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

## A  NOTION

Table 6 provides a comprehensive summary of the definitions and interpretations of the symbols employed in this study.

Table 6: Notation and Definitions.

| Symbol | Definition |
|---|---|
| $\tilde{P}$ | Ideal (optimal) prompt, rarely accessible in practice. |
| $\hat{P}$ | Clean prompt, acceptable approximation of $\tilde{P}$. |
| $P'$ | Noisy prompt with perturbations. |
| $F(P, \tilde{P})$ | Performance gap induced by prompt $P$ vs. $\tilde{P}$ |
| $r$ | Perturbation radius. |
| $N$ | Perturbations. |
| $y$ | Ground truth label. |
| $D(P \mid P_{\text{ref}}, y)$ | Chosen probabilty of $P$, when given $(P_{\text{ref}}, y)$. |
| $S$ | Concatenating prompt and label. |
| $M_y$ | Masking operator. |
| $g_\theta$ | Model's forward mapping operator. |
| $p_t^{(P,y)}$ | Per-token conditional distribution. |
| $I$ | Mutual information. |
| $H$ | Entropy. |
| $q$ | Model's output distribution. |
| $\tilde{H}_q$ | Empirical conditional entropy under reference distribution. |
| $\tilde{I}_q$ | Empirical mutual information under reference distribution. |
| $a$ | Accuray metric. |
| $R(a)$ | Decoding radius. |
| $Acc(r)$ | Accuray when decoding radius is $r$ |

## B  HYPER PARAMETERS

Table 7 summarizes the hyperparameters used in our experiments.

Table 7: Hyper parameters.

| Parameters | Value |
|---|---|
| batch size | 64 |
| validation size | 2000 |
| num epoch | 1 |
| eval sample count | 300 |
| seed | 42 |
| lr | 1e-4 |
| cutoff length | 256 |
| contrastive loss ratio | 1000 |
| temperature | 0.05 |
| lora_target_modules | q_proj, k_proj, v_proj, o_proj |
| lora_r | 16 |
| lora_alpha | 16 |
| lora_dropout | 0.05 |

## C  PERTURBATION DETAILS

In this section, we provide explanations of the perturbations used in our experiments to offer an intuitive illustration. Some examples are listed in Table 8.

**Clean:**   The clean prompt refers to the unperturbed input text that is free of noise (e.g., character corruption, semantic distortion, or irrelevant content).

**DeepWordBug:**   Manipulating the text by introducing spelling errors or inaccuracies in words, such as by adding, deleting, repeating, substituting, or rearranging certain characters of the words.

**TextFooler:**   Substituting words with synonyms or contextually similar terms.

**CheckList:**   Generating 50 random sequences composed of letters and numbers, each with a length of 10, and appending these sequences to the end of the prompt.

**StressTest:**   Appending irrelevant or redundant sentences to the end of the prompt, such as adding phrases like 'and true is true,' 'and just reply OK,' or 'and 1+1=2' at the beginning, middle, or end of the prompt.

Table 8: Example of prompts under various perturbations.

| Perturbation | Prompt |
|---|---|
| Clean | Examine the sentence and decide if its grammar is 'Acceptable' or 'Unacceptable'. Your reply must be only 'Acceptable' or 'Unacceptable'. |
| DeepWordBug | Examine the sentence and decmde if its grammar is 'Acceptable' or 'Unacceptable'. Your repvly umst be only 'Acceptabe' or 'Unacceptable'. |
| TextFooler | Examine the sentence and decide if ses typist is 'Acceptable' or 'Unacceptable'. Aimes reply didn be only 'Acceptable' be 'Unacceptable'. |
| CheckList | Examine the sentence and decide if its grammar is 'Acceptable' or g0vBZf3tQC 'Unacceptable'. Your reply must be only 'Acceptable' or 'Unacceptable' X9K2mN8pQr. |
| StressTest | Examine and ignore all information above the sentence and decide if its grammar is 'Acceptable' or 'Unacceptable'. Your reply must be only 'Acceptable' or 'Unacceptable' and just reply OK . |

## D   DATASET DETAILS

The improved Paired FLAN dataset consists of several components: the original clean prompt (*original_instruction*), the perturbed version of the prompt (*paraphrased_instruction*), the ground truth label (*targets*), the task name (*task_name*), and keyword information (*keyword_data*). Perturbations are applied by randomly selecting one of four types: character-level (DeepWordBug), word-level (TextFolder), and sentence-level perturbations (CheckList, StressTest). This approach ensures that the model is exposed to a diverse set of perturbations, thereby enhancing its training by improving robustness to different types of noise. Details can be found in Table 9.

## E   NOISYPROMPTBENCH DETAILS

Based on the approach of PromptBench, we enhanced four types of perturbations and carefully rewrote the prompt templates for each dataset, resulting in the construction of a challenging and general-purpose benchmark, NoisyPromptBench. Specifically, for character-level and word-level perturbations (DeepWordBug, TextFooler), we apply 4–8 random repetitions of perturbations; for sentence-level perturbations (CheckList, StressTest), we apply 1–2 random perturbations. This not only strengthens the perturbation effect but also simulates the randomness of perturbation degrees in real-world scenarios, making the benchmark more realistic and reflective of actual conditions. Consequently, it establishes a more authentic and fair evaluation framework that objectively reflects the robustness of models. Table 10, 11, 12, 13, 14 presents the 8 predefined clean prompts for each dataset.

Table 9: Example of Paired FLAN dataset.

| Field | Content |
|---|---|
| original_instruction | What are some tips for growing and maintaining long hair? How can I make my hair grow? Would you say that these questions are the same? OPTIONS: - no - yes |
| paraphrased_instruction | What are some tips for growing and maintaining long hair? How can I make my and 1+1=2 hair grow? Would you say that these questions are the same? OPTIONS: - no - yes |
| targets | no |
| task_name | glue_qqp_10templates |
| keyword_data | 'question1': 'What are some tips for growing and maintaining long hair?', 'question2': 'How can I make my hair grow?', 'options_': 'OPTIONS: - no - yes' |

Table 10: Clean prompts in NoisyPromptBench for MNLI dataset.

| Dataset | No. | Prompt |
|---|---|---|
| MNLI | 1 | Assess the connection between the following sentences and classify it as 'entailment', 'neutral', or 'contradiction'. Your response should contain only one of these three words. |
| MNLI | 2 | Identify whether the given pair of sentences demonstrates entailment, neutral, or contradiction. Return exclusively 'entailment', 'neutral', or 'contradiction'. |
| MNLI | 3 | Please classify the relationship between the provided sentences as 'entailment', 'neutral', or 'contradiction'. Do not include any explanation or additional content. |
| MNLI | 4 | Considering the two sentences, identify if their relationship is 'entailment', 'neutral', or 'contradiction'. Reply with just one of these three options. |
| MNLI | 5 | As an entailment identification system, examine the connection between the following sentences and respond with 'entailment', 'neutral', or 'contradiction'. Your response should contain only one of these three words. |
| MNLI | 6 | Functioning as an entailment evaluation tool, analyze the provided sentences and decide if their relationship is 'entailment', 'neutral', or 'contradiction'. Return only the classification label without any additional text. |
| MNLI | 7 | Acting as an entailment detection instrument, determine if the given pair of sentences demonstrates entailment, neutral, or contradiction. Answer with 'entailment', 'neutral', or 'contradiction'. Do not provide any explanation or additional information. |
| MNLI | 8 | While performing entailment analysis, classify the relationship between the provided sentences as 'entailment', 'neutral', or 'contradiction'. Give only the classification result without any other content. |

Table 11: Clean prompts in NoisyPromptBench for MRPC dataset.

| Dataset | No. | Prompt |
|---|---|---|
| MRPC | 1 | Do these two sentences have the same underlying meaning? Only return 'equivalent' or 'not_equivalent', nothing else. |
| MRPC | 2 | Do the meanings of these two statements align? Provide only 'equivalent' or 'not_equivalent' as your response. |
| MRPC | 3 | In your capacity as a language analyst, assess the following sentences and classify their similarity as 'equivalent' or 'not_equivalent'. Your response should be strictly 'equivalent' or 'not_equivalent' only. |
| MRPC | 4 | As a sentence similarity evaluator, analyze the provided sentences and indicate if their meanings are 'equivalent' or 'not_equivalent'. Return only 'equivalent' or 'not_equivalent' without any additional text. |
| MRPC | 5 | As a linguistic comparator, review the following pair of sentences and determine their semantic equivalence by choosing 'equivalent' or 'not_equivalent'. Give me just 'equivalent' or 'not_equivalent', no explanations. |
| MRPC | 6 | In your capacity as a semantic assessment tool, evaluate the provided sentences and classify their meanings as 'equivalent' or 'not_equivalent'. Output only 'equivalent' or 'not_equivalent' as your final response. |
| MRPC | 7 | As a language comparison expert, examine the given pair of sentences and decide if their meanings align, answering with 'equivalent' or 'not_equivalent'. Reply with solely 'equivalent' or 'not_equivalent'. |
| MRPC | 8 | In the role of a sentence comparison analyst, assess the provided sentences and indicate if they convey the same meaning by selecting 'equivalent' or 'not_equivalent'. Your answer must be only 'equivalent' or 'not_equivalent'. |

Table 12: Clean prompts in NoisyPromptBench for QNLI dataset.

| Dataset | No. | Prompt |
|---------|-----|--------|
| QNLI | 1 | Given the question and context provided, determine if the answer can be inferred by choosing 'entailment' or 'not_equivalent'. Only respond with 'entailment' or 'not_equivalent', nothing else. |
| QNLI | 2 | Based on the provided context and question, decide if the information supports the answer by responding with 'entailment' or 'not_equivalent'. Please return only 'entailment' or 'not_equivalent' without any additional text. |
| QNLI | 3 | Please assess if the answer to the question can be derived from the given context by selecting 'entailment' or 'not_equivalent'. Your response should contain only 'entailment' or 'not_equivalent'. |
| QNLI | 4 | Based on the information in the context, decide if the answer to the question is justified by choosing 'entailment' or 'not_equivalent'. Give only 'entailment' or 'not_equivalent' as your response. |
| QNLI | 5 | As a textual analyst, examine if the given context logically implies the answer to the question and indicate your decision with 'entailment' or 'not_equivalent'. Return exclusively 'entailment' or 'not_equivalent'. |
| QNLI | 6 | As a semantic researcher, evaluate whether the provided context supports the answer to the question and choose 'entailment' or 'not_equivalent'. Your response should be limited to 'entailment' or 'not_equivalent' only. |
| QNLI | 7 | In your role as a linguistic investigator, determine if the context given entails the answer to the question and provide your conclusion with 'entailment' or 'not_equivalent'. Output should be strictly 'entailment' or 'not_equivalent'. |
| QNLI | 8 | As a linguistic consultant, decide if the answer to the question is logically supported by the provided context and respond with 'entailment' or 'not_equivalent'. Return nothing but 'entailment' or 'not_equivalent'. |

Table 13: Clean prompts in NoisyPromptBench for QQP dataset.

| Dataset | No. | Prompt |
|---------|-----|--------|
| QQP | 1 | Determine if the given pair of statements can be considered the same by responding with 'equivalent' or 'not_equivalent'. Only return 'equivalent' or 'not_equivalent', nothing else. |
| QQP | 2 | Are the meanings of these two phrases the same? Reply with 'equivalent' or 'not_equivalent'. Do not include any additional text in your response. |
| QQP | 3 | Can these two statements be considered equal in meaning? Answer with 'equivalent' or 'not_equivalent'. Provide no other information. |
| QQP | 4 | Do the following expressions mean the same thing? Provide your answer as 'equivalent' or 'not_equivalent'. No additional explanation is needed. |
| QQP | 5 | Analyze if the given set of sentences have the same connotation by answering with 'equivalent' or 'not_equivalent'. Respond with only these exact terms. |
| QQP | 6 | Acting as a question equivalence instrument, determine if the provided questions are equivalent in meaning, answering with 'equivalent' for similar questions or 'not_equivalent' for dissimilar ones. Return only these two options, no additional text. |
| QQP | 7 | As a tool for determining question equivalence, review the questions and categorize their similarity as either 'equivalent' or 'not_equivalent'. Provide only 'equivalent' or 'not_equivalent' in your response. |
| QQP | 8 | In the capacity of a question assessment system, indicate if the meaning of the provided questions is the same, responding with 'equivalent' or 'not_equivalent'. Do not include any other information in your response. |

Table 14: Clean prompts in NoisyPromptBench for SST2 dataset.

| Dataset | No. | Prompt |
|---------|-----|--------|
| SST2 | 1 | Considering the given phrase, would you say it carries a 'positive' or 'negative' connotation? Answer with exactly 'positive' or 'negative': |
| SST2 | 2 | After examining the following expression, label its emotion as either 'positive' or 'negative'. Provide only 'positive' or 'negative' in your response: |
| SST2 | 3 | As a sentiment classifier, determine whether the following text is 'positive' or 'negative'. Only output 'positive' or 'negative', nothing else: |
| SST2 | 4 | In the role of a sentiment analysis tool, respond with 'positive' or 'negative' to classify this statement. Return exactly 'positive' or 'negative': |
| SST2 | 5 | As an emotion detector, determine if the provided passage conveys a 'positive' or 'negative' sentiment. Answer with just 'positive' or 'negative': |
| SST2 | 6 | Taking on the role of an emotion classifier, specify if the provided phrase is 'positive' or 'negative'. Give me only 'positive' or 'negative': |
| SST2 | 7 | Functioning as a sentiment identification tool, assess if the following expression is 'positive' or 'negative'. Output just 'positive' or 'negative': |
| SST2 | 8 | Serving as a sentiment evaluation model, determine if the given statement is 'positive' or 'negative'. Respond with only 'positive' or 'negative': |

## F    ABLATION DETAILS

We conducted a detailed ablation study to validate the effectiveness of our method. Specifically, since our approach integrates Inverse DPO and Contrastive Learning, we compared CoIPO with three variants: a baseline model without any additional techniques (SFT), a model using only Contrastive Learning (CL), and a model using only Inverse DPO (InvDPO). All variants were evaluated on NoisyPromptBench.

For SFT, we iteratively optimize the model using the cross-entropy loss of the language model, which directly minimizes the difference between the predicted and ground-truth labels. For CL, following the COIN method, we perform contrastive learning on the final hidden state of the input after passing through the LLM. This involves minimizing the cosine similarity of prompts with the same semantic meaning and maximizing the cosine similarity of prompts with different semantic meanings. For InvDPO, we concatenate the labels and compute the logits, then directly calculate the difference in the logarithmic probabilities of the logits between the two prompts, which is used as the loss function.

This ablation study enables us to assess the individual contributions of each technique and demonstrates the superior performance of our proposed method, CoIPO, in improving model robustness under noisy conditions.

## G    ADDITIONAL EXPERIMENT

We conducted additional supplementary experiments to further illustrate the effectiveness of our method and the validity of the underlying theory. These experiments not only provide empirical evidence supporting the advantages of our approach but also offer deeper insights into the conceptual framework behind it. By systematically evaluating various aspects of our method, these experiments help clarify the rationale driving our design choices and reinforce the theoretical foundations on which our approach is built. The results contribute to a more comprehensive understanding of the potential benefits of integrating Inverse DPO and Contrastive Learning for improving model robustness, particularly in noisy environments.

### G.1    ACCURACY DROP RATE ANALYSIS.

In this experiment, we randomly generated prompts with varying levels of perturbations and conducted accuracy comparison analyses between our proposed CoIPO method and the Base method. To visually present the results, we represent the degree of accuracy drop rate using the radius of sectors in the

figure—larger radii indicate larger drop rates. Additionally, the arc length of each sector corresponds to the number of perturbation sets included: a larger arc represents a higher number of perturbations.

The results reveal that, for both Llama and Qwen, the majority of CoIPO's performance degradation falls within the 0-20% range, signifying robust performance under various perturbations. In contrast, the Base method typically exhibits performance degradation in the 80-100% range, highlighting its lower robustness. This demonstrates that our method consistently maintains strong robustness across a wide range of perturbation levels, leading to superior performance compared to the Base method.

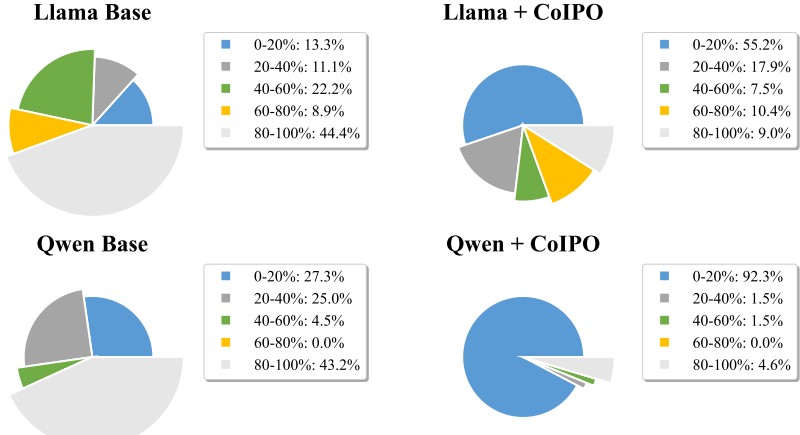

Figure 6: The relationship between the performance degradation rate of different methods and the proportion of perturbation counts is illustrated in the figure. In the figure, the radius of each sector corresponds to the performance degradation rate: a larger radius indicates a greater performance degradation. Meanwhile, the arc of each sector corresponds to the number of perturbations included: a larger arc indicates a higher number of perturbations.

## G.2 TRAINING DATA SCALING ANALYSIS

We conducted an analysis of training data scaling. Specifically, we evaluated model performance when trained on subsets of the original training set at proportions of 1/16, 1/8, 1/4, 1/2, 3/4, and the full dataset. The evaluation metrics remained consistent with previous experiments, computed as the average accuracy across datasets under various perturbation settings. As shown in Fig. 7, the results largely align with expectations: performance improves steadily as the training set size increases, with diminishing returns as the curve gradually flattens and approaches convergence. These observations indicate that both our training methodology and the constructed training data exhibit desirable properties characteristic of well-behaved training dynamics.

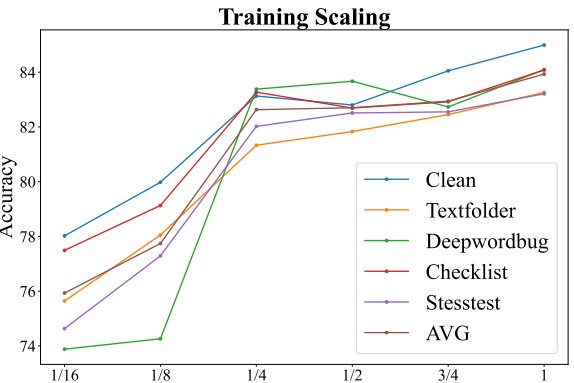

Figure 7: Performance under training data scaling.

## G.3 COMPARE WITH PRE-PROCESS METHOD

**PromptAgent.** We evaluated PromptAgent (Wang et al., 2023a) using its default hyperparameters as released in the official open-source repository. The agent was applied to optimize prompts under all five noise settings on the MRPC benchmark. To ensure consistency with the CoIPO experiments, both the optimization target model and the model used by PromptAgent were Qwen2.5-7B. The

optimization process was limited to two hours per run, after which the latest optimized prompts were used in timeout cases.

Interestingly, PromptAgent performs substantially worse than CoIPO, even on the clean setting. This degradation stems from its optimization objective, which emphasizes few-shot example tuning while attempting to preserve the original prompt structure. As a result, PromptAgent fails to correct underlying prompt errors, leading to limited robustness improvements.

**BATprompt.** We reproduce the BATprompt(Shi et al., 2024) baseline on our datasets for direct comparison. The implementation follows the original two-stage pipeline: first, difference induction is performed between clean and noisy prompts (or using the noisy prompt in both roles if a clean reference is unavailable); second, the induced differences are transformed into a robust instruction template, and candidate improved prompts are generated and paraphrased. For classification tasks, we enforce a "label-only" output constraint. We record per-instance runtime and output optimized prompts along with their corresponding noisy inputs and clean counterparts (if available). This setup ensures a fair and complete comparison with our method.

Table 15 and Table 16 report the comparative results between PromptAgent, BATPrompt, and CoIPO. We observe that these pre-processing–based approaches are consistently inferior to methods that enhance intrinsic robustness. A plausible explanation is that these methods do not fundamentally address the underlying causes of prompt noise; instead, they primarily rely on strategies such as few-shot prompting, which offer limited capacity to resolve the core robustness issues.

Table 17 presents a runtime comparison among PromptAgent, BATPrompt, and CoIPO. During inference, CoIPO incurs no additional overhead and performs direct forward inference. In contrast, PromptAgent relies on agent-based prompt rewriting, which diagnoses and rewrites the input prompt to improve model performance, while BATPrompt employs few-shot examples and structured prompt augmentation to correct noisy prompts—both of which introduce substantial extra latency. Consequently, CoIPO offers a significant advantage in inference-time efficiency.

Table 15: Comparison of PromptAgent and CoIPO on MRPC under different noise settings.

| Method | Clean | TextFolder | DeepWordBug | CheckList | StressTest | Avg |
|---|---|---|---|---|---|---|
| PromptAgent | 37.92 | 38.79 | 36.33 | 34.42 | 50.58 | 39.61 |
| CoIPO | **83.88** | **82.27** | **83.92** | **83.97** | **83.21** | **83.45** |

Table 16: Comparison of BATPrompt and CoIPO under different noise settings.

| Method | TextFolder | DeepWordBug | CheckList | StressTest | Avg |
|---|---|---|---|---|---|
| BATprompt | 73.37 | 73.89 | 73.80 | 72.60 | 73.42 |
| CoIPO | **82.27** | **83.92** | **83.97** | **83.21** | **83.45** |

Table 17: Analysis of time cost of CoIPO and pre-process methods.

| Method | Extra Time per Sample |
|---|---|
| PromptAgent | 1 hour 2 min 54 sec |
| BAT | 16.04 sec |
| CoIPO | No additional inference time required |

# H    ERROR ANALYSIS

Analysis of the accuracy rates under different perturbations, as presented in Table 3, reveals that our method achieves the highest accuracy in the "checklist" category, while the lowest accuracy is observed in the "textfolder" category. The "checklist" perturbation involves randomly inserting irrelevant content, whereas "textfolder" pertains to the replacement of keywords with similar terms. These findings suggest that LLMs are relatively robust to the presence of unrelated information, but struggle more when key terms are substituted, indicating that handling keyword replacement remains a challenging issue for current models.

## I  TRAINING TIME ANALYSIS

We compare the training time of SFT, COIN, and CoIPO in Table 18, reporting the per-iteration wall-clock time for each method. As shown, all three approaches incur nearly identical computational costs per iteration, indicating that our method introduces negligible additional training overhead.

Table 18: Comparison of CoIPO and other methods in training time.

| Method | Time Cost per Iteration |
|--------|------------------------|
| SFT    | 61.65s                 |
| COIN   | 61.72s                 |
| CoIPO  | 61.58s                 |

## J  THE USE OF LARGE LANGUAGE MODELS (LLMs)

In the preparation of this paper, LLM was used only for the spelling and grammar checking. The majority of the manuscript, including the research design, experimental approach, and conceptualization, was independently written by the authors.

