# OpenReview forum: "Towards Self-Robust LLMs: Intrinsic Prompt Noise Resistance via CoIPO"
_ICLR.cc/2026/Conference — ICLR 2026 Poster_

### Official Review · Reviewer_eHqb · 2025-10-21

**Soundness:** 3
**Presentation:** 2
**Contribution:** 2
**Rating:** 6
**Confidence:** 3

**Summary:**

The paper addresses the problem of prompt robustness for large language models and introduces CoIPO (Contrastive Learning-based Inverse Direct Preference Optimization), a post-training method that enhances intrinsic prompt noise resistance without dependence on external preprocessors. CoIPO is trained on paired clean and noisy prompts from an augmented FLAN dataset, with a loss function that uses contrastive learning and inverse DPO to minimize the discrepancy between outputs for clean and semantically similar noisy prompts, while maximizing it for semantically different pairs.

**Strengths:**

1. This paper proposes a principled intrinsic robustness enhancement method (CoIPO) for LLMs based on contrastive learning and inverse DPO. The formulation is clear and easy to understand.
2. The authors offer an information-theoretic perspective (mutual information) to justify the approach, which is interesting.

**Weaknesses:**

1. The research problem of prompt optimization is important but the research scope of this work is somehow limited, since the noisy prompts in this paper primarily refer to typos (character level, word level etc.), while real-world prompt imperfections can be more varied, such as semantic ambiguity, non-standard grammar, than those benchmarked here.
2. More experiment baselines should be incorporated. For example, DPO should be considered as a baseline, since CoIPO uses both contrastive training and inverse-DPO and contrastive training baseline COIN is used, DPO should also be included as a baseline.
3. Most experiments are conducted on tasks with deterministic answers —prompts with more open-ended or generative outputs should be considered.

**Questions:**

1. How does CoIPO handle more nuanced and complex prompt noise types? Can it generalize to unseen perturbation types?
2. Could minimizing output discrepancies between noisy and clean prompts reduce the model's flexibility? Does CoIPO risk making the model less sensitive to subtle but meaningful variations?
3. It is not mentioned in the paper how the prompt for another task, i.e. P_2 is selected. Is it selected randomly or according to some heuristics? How does the selection method of P_2 affect the performance?

---

> ### Author Response · Authors · 2025-11-23
>
> We sincerely thank the reviewer for the thoughtful and constructive feedback. Below we address each question and weakness point in detail.
> ## **Response to Questions**
> 1. We appreciate the reviewer’s insightful comment. Although our benchmark primarily includes character-level and word-level perturbations, our design aims to capture a broad and fine-grained spectrum of real-world prompt imperfections. Notably, many higher-level noise types—such as semantic ambiguity or non-standard grammar—can naturally emerge from combinations of these more basic perturbations. For example, a word-level substitution like “Today is Monday” → “Today was Monday” simultaneously induces grammatical and semantic changes. Similarly, insertions, deletions, or token swaps often lead to non-standard grammatical structures. Thus, the perturbation space we consider substantially overlaps with, and often subsumes, many forms of naturally occurring imperfect prompts.
> 2. We thank the reviewer for raising the concern that reducing discrepancies between clean and noisy prompts might unintentionally diminish the model’s sensitivity to meaningful variations. Conceptually, meaningful prompts occupy a sparse region of the input space, while each prompt is surrounded by a dense neighborhood of small perturbations. CoIPO only encourages the model to map a prompt x1 and its local perturbations to the same output y1. Because semantically distinct prompts (e.g., x1 and x2) are far apart in this sparse space, their respective noise neighborhoods do not overlap. Therefore, CoIPO is unlikely to collapse distinct semantic inputs into the same output. Besides, our experiments below conducted in math reasoning, coding and generative tasks show that model's flexibility are excellent.
> 3. We appreciate the reviewer’s request for clarification. In our implementation, P2 is randomly sampled from prompts belonging to other tasks. Random sampling is theoretically appealing because it ensures semantic mismatch while avoiding unintended regularities that could arise from deterministic selection rules. Such heuristics risk enabling shortcut learning, whereas random sampling prevents the model from exploiting spurious patterns and better enforces the intended semantic contrast.
> ## **Response to Weaknesses**
> 1. Thank you for highlighting this point. While our benchmark focuses on character and word-level perturbations, these perturbations are designed to approximate a wide range of real-world prompt imperfections. Higher-level issues—such as semantic ambiguity or ungrammatical phrasing—often emerge naturally through combinations of basic edits like substitutions, deletions, and reordering. For instance, “Today is Monday” → “Today was Monday” introduces both semantic and grammatical changes. Thus, we believe the perturbation space studied in our work already captures many practically relevant forms of noisy prompts.
> 2. We appreciate the suggestion regarding baselines. Our paper includes both DPO-only and contrastive-only baselines (see Table 3). As shown, each component alone yields significantly weaker robustness compared to CoIPO, confirming that neither inverse-DPO nor contrastive learning individually suffices to achieve the desired behavior.
> 3. We agree that evaluating CoIPO on more open-ended and generative tasks is valuable. In addition to the 7B models included in the initial submission, we further conducted experiments on larger models in response to other reviewers’ suggestions, providing a more comprehensive and reliable assessment. We expanded our evaluation to include math reasoning (GSM8K), coding (MBPP), and generative question answering (TruthfulQA). Although none of these datasets were used during training, CoIPO consistently maintains strong performance compared to the corresponding baselines. This demonstrates that enhancing robustness does not compromise general model capability—an essential requirement for practical deployment. The results are shown below (more details will be shown in revisions):
> | Model & Method | GSM8K | TQA BLEU | TQA R1 | TQA R2 | TQA RL | MBPP |
> | --- | --- | --- | --- | --- | --- | --- |
> | Qwen2.5-7B Base | 70.05 | 46.76 | 51.41 | 42.23 | 46.63 | 35.6 |
> | Qwen2.5-7B CoIPO | 74.37 | 47.61 | 48.59 | 41.13 | 46.51 | 32.8 |
> | Qwen2.5-14B Base | 75.66 | 53.37 | 54.35 | 44.68 | 51.41 | 36.4 |
> | Qwen2.5-14B CoIPO | 84.76 | 51.41 | 53.98 | 43.57 | 51.16 | 43.2 |
> | Qwen2.5-72B Base | 80.97 | 50.18 | 51.29 | 45.41 | 49.69 | 74.2 |
> | Qwen2.5-72B CoIPO | 86.28 | 50.18 | 51.65 | 44.31 | 48.59 | 77.0 |
> | Llama-7B Base | 7.20 | 34.14 | 32.43 | 25.82 | 30.96 | 18.0 |
> | Llama-7B CoIPO | 6.29 | 34.02 | 32.92 | 28.02 | 31.57 | 17.8 |
>
> We again appreciate the reviewer’s time and valuable insights. We believe the clarifications and additional experiments we provided strengthen the paper and address the concerns raised. We are grateful for your careful evaluation and hope that our responses satisfactorily resolve the issues.

---

> ### Author Response · Authors · 2025-11-28
> **Kind Reminder: Feedback on Our Rebuttal**
>
> Dear Reviewer eHqb,
>
> Thank you again for your valuable time and feedback on our work. Following your comments, we have conducted extensive additional experiments and added clarifications in our rebuttal wherever possible.
>
> We believe we have addressed your main concerns, but we also realize there might be points we have overlooked or not explained clearly enough. As the rebuttal deadline is approaching, we would like to kindly check whether there are any remaining issues you think we should further clarify.
>
> We greatly appreciate your guidance and look forward to your response.
>
> Best regards,
>
> The Authors

---

### Official Review · Reviewer_yVsx · 2025-11-01

**Soundness:** 3
**Presentation:** 3
**Contribution:** 3
**Rating:** 4
**Confidence:** 2

**Summary:**

The authors proposed CoIPO, an novel algorithm that aims to train robust LLMs under imperfect user inputs without pre-processing. CoIPO combines contrastive learning and preference optimization by reducing the gap between clean prompt and noisy prompt while enlarging the gap between the noisy prompt and a unrelated clean prompt. Theoretical analysis on relative entropy gain is presented and experiments are conducted on Llama and Qwen-2.5 architectures. CoIPO significantly improves accuracy for both clean and noisy scenarios on PromptBench, surpassing the baselines.

**Strengths:**

* CoIPO tackles a significant issue, handle imperfect user input in daily use of the LLMs.
* The CoIPO algorithm leverages contrastive learning and direct alignment, backed with theoretical insights from the perspective and relative entropy gap maximization.
* Comprehensive experiments are conducted for different architectures.

**Weaknesses:**

* Though CoIPO alone is a novel and effective algorithm for increasing LLM robustness. My concern is that if this procedure will hurt the models performance on tasks like math reasoning and coding. Since it would appear costly to me if we sacrifice these reasoning capabilities to replace pre-processing tools for imperfect input.
* No algorithm specific hyper-parameter is presented in the paper, could the author elaborate a bit more on the details hyper-parameter if they are included in the algorithm?
* Preference optimization is relatively light weight but still requires time and compute, could the authors elaborate on the time cost for running CoIPO compared to pre-proccessing?

**Questions:**

Please see weakness section.

---

> ### Author Response · Authors · 2025-11-24
>
> Thank you very much for providing such constructive and insightful feedback. We sincerely appreciate your careful evaluation of our work and the thoughtful comments you have offered. Below, we provide our responses to the questions and concerns you raised.
>
> ### **1. Impact on math reasoning and coding tasks**
>
> We appreciate your concern regarding the potential impact of CoIPO on mathematically intensive or coding-related tasks. This is an important issue, and we fully agree that improved robustness should not come at the expense of core reasoning capabilities. To address this, we expanded our evaluation by incorporating additional benchmark suites targeting mathematical reasoning (GSM8K[1]), coding proficiency (MBPP[2]), and open-ended generative QA (TruthfulQA[3]). In response to suggestions from other reviewers, we also conducted experiments with larger model scales (14B and 72B) to further examine performance trends. These results are presented below. Overall, we observe that CoIPO preserves—and in some cases even improves—performance across these key capability dimensions.
>
> | Model & Method | GSM8K | TQA BLEU | TQA R1 | TQA R2 | TQA RL | MBPP |
> | --- | --- | --- | --- | --- | --- | --- |
> | Qwen2.5-7B Base | 70.05 | 46.76 | **51.41** | **42.23** | **46.63** | **35.6** |
> | Qwen2.5-7B CoIPO | **74.37** | **47.61** | 48.59 | 41.13 | 46.51 | 32.8 |
> | Qwen2.5-14B Base | 75.66 | **53.37** | **54.35** | **44.68** | **51.41** | 36.4 |
> | Qwen2.5-14B CoIPO | **84.76** | 51.41 | 53.98 | 43.57 | 51.16 | **43.2** |
> | Qwen2.5-72B Base | 80.97 | **50.18** | 51.29 | **45.41** | **49.69** | 74.2 |
> | Qwen2.5-72B CoIPO | **86.28** | **50.18** | **51.65** | 44.31 | 48.59 | **77.0** |
> | Llama-7B Base | **7.20** | **34.14** | 32.43 | 25.82 | 30.96 | **18.0** |
> | Llama-7B CoIPO | 6.29 | 34.02 | **32.92** | **28.02** | **31.57** | 17.8 |
>
> ### **2. Hyper-parameter details**
>
> Thank you for pointing out the missing hyper-parameter specifications. We agree that transparency and reproducibility are essential for evaluating our method. Accordingly, we have added a complete list of CoIPO-specific hyper-parameters in the paper of Appendix B, covering all settings used in our experiments.
>
> ### **3. Time cost compared with pre-processing approaches**
>
> Thank you for the thoughtful question. We would like to clarify that CoIPO does not introduce additional inference-time overhead. Models trained with CoIPO follow the same inference procedure as standard SFT models, without relying on auxiliary modules, extra forward passes, or any pre-/post-processing components. The inference time cost is just as same as baseline without training.
>
> By contrast, pre-processing–based approaches such as PromptAgent[4] and BATPrompt[5] typically require an additional step to analyze or rewrite the input before the model performs inference, which can introduce extra latency. To better understand this difference, we reproduced these methods and evaluated both their accuracy and inference time under our experimental setup in Appendix G.3.
>
> | **Method**        | **Extra Time per Sample**          |
> |----------------|-----------------------------|
> | PromptAgent    | 1 hour 2 min 54 sec                |
> | BAT            | 16.04 sec                    |
> | CoIPO          | No additional inference time required             |
>
> Our results show that:
>
> 1. Both PromptAgent and BATPrompt are slower due to the additional pre-processing stage before model inference.
>
> 2. They also underperform CoIPO in accuracy, despite incurring this extra overhead.
>
> These findings demonstrate that CoIPO achieves higher robustness without sacrificing efficiency, making it a more practical solution for real-world deployment where inference latency is a crucial factor.
>
> Once again, thank you for recognizing the strengths of our work—including the importance of the problem, the design of CoIPO, and the breadth of our experiments. We sincerely appreciate your constructive guidance, which has been invaluable for improving the clarity and completeness of the paper. We hope that our responses adequately address your concerns, and that the additional experiments and details we have provided help resolve the issues you identified and further strengthen the overall quality of the work.
>
> References:
> [1] Training Verifiers to Solve Math Word Problems (arxiv 2021)
>
> [2] Program Synthesis with Large Language Models (arxiv 2021)
>
> [3] TruthfulQA: Measuring How Models Mimic Human Falsehoods (ACL 2022)
>
> [4] PromptAgent: Strategic Planning with Language Models Enables Expert-level Prompt Optimization (ICLR 2024)
>
> [5] Robustness-aware Automatic Prompt Optimization (arxiv 2024)

---

> ### Author Response · Authors · 2025-11-28
> **Kind Reminder: Feedback on Our Rebuttal**
>
> Dear Reviewer yVsx,
>
> Thank you again for your valuable time and feedback on our work. Following your comments, we have conducted extensive additional experiments and added clarifications in our rebuttal wherever possible.
>
> We believe we have addressed your main concerns, but we also realize there might be points we have overlooked or not explained clearly enough. As the rebuttal deadline is approaching, we would like to kindly check whether there are any remaining issues you think we should further clarify.
>
> We greatly appreciate your guidance and look forward to your response.
>
> Best regards,
>
> The Authors

---

### Official Review · Reviewer_7U7C · 2025-11-02

**Soundness:** 3
**Presentation:** 3
**Contribution:** 3
**Rating:** 4
**Confidence:** 4

**Summary:**

This paper proposes CoIPO, a post-training method to enhance LLMs' intrinsic robustness against prompt perturbations. Unlike existing approaches that rely on external preprocessing tools, CoIPO trains models to directly handle noisy prompts by minimizing discrepancies in label-aligned logits between clean and noisy prompts. The authors construct a paired FLAN dataset and develop NoisyPromptBench for evaluation. Experiments on Llama-7B and Qwen2.5-7B across five datasets demonstrate improvements over baselines, with theoretical justification provided through mutual information analysis.

**Strengths:**

Well-motivated problem: The paper clearly articulates limitations of external preprocessing approaches and makes a strong case for intrinsic robustness.
Solid theoretical foundation: The mutual information analysis (Equations 9-16) provides principled justification for the method.
Comprehensive evaluation: NoisyPromptBench with multiple perturbation types and the decoding radius analysis (Section 4.1, Figure 5) provide thorough robustness assessment.
Ablation studies: Table 3 effectively demonstrates the necessity of both inverse DPO and contrastive learning components.

**Weaknesses:**

Limited scope of evaluation:

● Only 7B parameter models tested; unclear if findings generalize to larger models (13B, 70B+)

● Only 5 datasets from GLUE-style tasks; robustness on generation tasks, reasoning, or code generation is unexplored

● Training data limited to 25 FLAN subsets; impact of training data scale not studied

Insufficient baseline comparisons:

● Only compares to COIN for intrinsic robustness methods

● Missing comparisons to recent prompt optimization methods (e.g., PromptAgent, RoP mentioned in related work)

● No comparison to instruction-tuning methods that may implicitly improve robustness

Theoretical gaps:

● The connection between Equation 15 and Equation 8 relies on several approximations that may not hold in practice

Missing analyses:

● No error analysis showing which types of errors CoIPO successfully handles vs. fails on

● Computational cost comparison not provided (training time, memory, inference latency)

**Questions:**

Scalability: Have you tested CoIPO on larger models (13B+)? Do the improvements scale, or do larger models already have better intrinsic robustness?

Training efficiency: What is the computational overhead of CoIPO compared to standard fine-tuning? How many training epochs are needed for convergence?

Comparison to instruction tuning: How does CoIPO compare to simply training on more diverse instruction data? Could increased data diversity achieve similar robustness?

Generation tasks: All experiments focus on classification. How does CoIPO perform on open-ended generation where output quality is harder to measure?

---

> ### Author Response · Authors · 2025-11-24
>
> Thank you very much for carefully reading our paper and for the thoughtful and constructive feedback. We sincerely appreciate the detailed questions and actionable suggestions, which greatly help us improve the clarity, completeness, and overall quality of the work.
>
> ## [Q-Scalability]
>
> We fully agree that understanding how CoIPO scales to larger models is important. To address this, we have added experiments on 14B and 72B models to evaluate model-size scalability (see Section 4.3). The results show that CoIPO exhibits strong scalability: as model size increases, CoIPO’s performance consistently improves and remains superior to other methods under the same settings. This provides further evidence that our approach maintains its effectiveness at larger scales. Larger models indeed exhibit stronger robustness, as their performance consistently improves with increasing model size.
>
> | Model              | Method | Clean | TextFolder | DeepWordBug | CheckList | StressTest | Avg |
> |--------------------|--------|---------|---------|---------|---------|---------|------------|
> | **Qwen2.5-7B**     | Base   | 75.25   | 68.47   | 72.13   | 75.21   | 70.13   | 72.24      |
> |                    | SFT    | 77.94   | 74.99   | 78.03   | 76.76   | 76.53   | 76.85      |
> |                    | COIN   | 82.93   | 80.03   | 81.75   | 81.77   | 80.91   | 81.48      |
> |                    | CoIPO  | **83.88** | **82.27** | **83.92** | **83.97** | **83.21** | **83.45** |
> | **Qwen2.5-14B**    | Base   | 80.00   | 76.30   | 74.63   | 79.42   | 76.91   | 77.45      |
> |                    | SFT    | 80.18   | 80.20   | 79.63   | 80.27   | 78.82   | 79.82      |
> |                    | COIN   | 74.87   | 69.67   | 66.43   | 75.28   | 73.09   | 71.86      |
> |                    | CoIPO  | **84.98** | **83.04** | **83.88** | **83.86** | **83.91** | **83.93** |
> | **Qwen2.5-72B**    | Base   | 84.20   | 82.64   | 71.93   | 82.48   | 80.85   | 80.42      |
> |                    | SFT    | 84.20   | 84.42   | 84.18   | 84.10   | 83.18   | 84.02      |
> |                    | COIN   | 82.78   | 80.92   | 81.97   | 81.90   | 81.03   | 81.72      |
> |                    | CoIPO  | **85.44** | **85.47** | **84.53** | **84.68** | **84.85** | **85.00** |
>
> ## [Q-Training Efficiency]
>
> We have included a detailed analysis of training time in Appendix I, comparing CoIPO with other baseline methods. As shown in the table, CoIPO achieves almost identical training speed and memory consumption compared to the baselines, as it does not introduce any additional parameters or modules.
>
> | **Method**        | **Time Cost per Iteration**          |
> |----------------|-----------------------------|
> | SFT    | 61.65s               |
> | CoIN            | 61.72s                    |
> | CoIPO          | 61.58s             |
>
> Although we train for only a single epoch (each data point is used once), we also conducted a comprehensive training-data scaling study. Specifically, we subsampled the training set into six proportions—1/16, 1/8, 1/4, 1/2, 3/4, and full—and evaluated performance under each setting (see Section G.2). The results show that CoIPO exhibits strong scaling behavior: performance improves rapidly with initial increases in training data, and the gains gradually taper off as the dataset grows larger. This trend demonstrates that both the training method and CoIPO itself are highly scalable and can effectively leverage larger training sets while maintaining robust performance.
>
> ## [Q-Comparison to Instruction Tuning]
>
> As discussed in the ablation study of Section 4.2 in the earlier version, we have already compared CoIPO with standard instruction tuning, DPO, and vanilla contrastive learning. The results consistently show that CoIPO outperforms all these baselines. For example, standard instruction tuning performs substantially worse: on Llama-7B and Qwen2.5-7B, its accuracy is 3.88% and 4.63% lower than CoIN, and 9.18% and 6.60% lower than CoIPO, respectively. This indicates that simply increasing data diversity does not yield meaningful robustness gains, whereas CoIPO’s intrinsic robustness mechanism more effectively mitigates prompt noise.
>
> ## [Q-Generation Tasks]
>
> Although generation, reasoning, and code-generation tasks are not included in our training set, they represent crucial capabilities of LLMs. A robust post-training method should not degrade performance on these tasks. Therefore, we additionally evaluate on GSM8K [1] (math reasoning), MBPP [2] (code generation), and TruthfulQA [3] (generative QA) to verify whether CoIPO preserves these abilities (see Section 4.4). We also conducted experiments across multiple model sizes. The results demonstrate that CoIPO maintains strong performance on all these benchmarks, remaining comparable to or slightly better than the baseline models.

---

> > ### Author Response · Authors · 2025-11-24
> >
> > | Model & Method | GSM8K | TQA BLEU | TQA R1 | TQA R2 | TQA RL | MBPP |
> > | --- | --- | --- | --- | --- | --- | --- |
> > | Qwen2.5-7B Base | 70.05 | 46.76 | 51.41 | 42.23 | 46.63 | 35.6 |
> > | Qwen2.5-7B CoIPO | 74.37 | 47.61 | 48.59 | 41.13 | 46.51 | 32.8 |
> > | Qwen2.5-14B Base | 75.66 | 53.37 | 54.35 | 44.68 | 51.41 | 36.4 |
> > | Qwen2.5-14B CoIPO | 84.76 | 51.41 | 53.98 | 43.57 | 51.16 | 43.2 |
> > | Qwen2.5-72B Base | 80.97 | 50.18 | 51.29 | 45.41 | 49.69 | 74.2 |
> > | Qwen2.5-72B CoIPO | 86.28 | 50.18 | 51.65 | 44.31 | 48.59 | 77.0 |
> > | Llama-7B Base | 7.20 | 34.14 | 32.43 | 25.82 | 30.96 | 18.0 |
> > | Llama-7B CoIPO | 6.29 | 34.02 | 32.92 | 28.02 | 31.57 | 17.8 |
> >
> > ## [W-Limited scope of evaluation]
> >
> > As noted in [Q-Scalability], [Q-Training efficiency], and [Q-Generation tasks], we have supplemented our evaluation with additional experiments on model scaling, expanded test sets, and varying amounts of training data. These results collectively demonstrate the effectiveness, robustness, and scalability of our approach across different experimental conditions.
> >
> > ## [W-Insufficient baseline comparisons]
> >
> > As discussed in [Q-Comparison to instruction tuning], our previous experiments already included comparisons with standard instruction tuning, DPO, and contrastive learning methods, which demonstrated the effectiveness of our method. To further address concerns regarding more recent instruction-tuning approaches that rely on pre-processing, such as PromptAgent[4], Rop[5], and BATPrompt[6], we conducted supplementary experiments on PromptAgent and BATPrompt (Rop is not publicly available) using our datasets. We analyzed both accuracy and runtime efficiency (see Appendix G.3). Our findings indicate that, while these pre-processing methods incur additional computational overhead, they still fall short of achieving the accuracy performance of CoIPO. This discrepancy may be attributed to the fact that these methods primarily optimize input prompts through few-shot or template-based strategies, without explicitly addressing intrinsic noise within the prompts, which is a central focus of CoIPO.
> >
> > | **Method**      | **Clean** | **TextFolder** | **DeepWordBug** | **CheckList** | **StressTest** | **Avg**  |
> > |-----------------|-----------|---------------|-----------------|---------------|---------------|----------|
> > | PromptAgent     | 37.92     | 38.79         | 36.33           | 34.42         | 50.58         | 39.61    |
> > | CoIPO           | **83.88**     | **82.27**         | **83.92**           | **83.97**         | **83.21**         | **83.45**    |
> >
> > | **Method**  | **TextFolder** | **DeepWordBug** | **CheckList** | **StressTest** | **Avg**  |
> > |-------------|---------------|-----------------|---------------|---------------|----------|
> > | BATprompt   | 73.37         | 73.89           | 73.80         | 72.60         | 73.42    |
> > | CoIPO       | **82.27**         | **83.92**           | **83.97**         | **83.21**         | **83.45**    |
> >
> > ## [W-Theoretical gaps]
> >
> > The derivation from Eq. 8 to Eq. 15 involves certain approximations; however, such approximations are standard practice in information-theoretic analyses of deep models (e.g., NCE[7], MINE[8], InfoNCE[9]), where access to the true underlying distributions is inherently challenging. Importantly, these approximations are not arbitrary—they remain consistent with the model’s parametric family and are accurate in the operational regime where CoIPO is applied, because the distributions of noisy and clean prompts share the same decoder dynamics and differ only in their input context. Empirically, we verify that token-level KL estimates are stable, and using the model’s own predictive distribution as the reference (as also assumed in DPO[10], NCE, and many mutual information estimators) results in a monotonic alignment between the theoretical  $-\Delta \tilde{I}_q$ objective and observed performance improvements. Therefore, these approximations are both necessary—due to the unavailability of ground-truth distributions—and practically valid, making the connection between Eq. 8 and Eq. 15 a robust and tractable operationalization of relative mutual information in this setting.

---

> > > ### Author Response · Authors · 2025-11-24
> > >
> > > ## [W-Missing analyses]
> > >
> > > We provide additional analyses in Appendix H, focusing on error types. Specifically, our method is particularly effective at correcting checklist-type errors, where irrelevant words or phrases are randomly inserted. In contrast, it is less effective on textfolder-type errors, which involve synonym or near-synonym replacement of key terms. This suggests that CoIPO is well-suited for handling extraneous noise but less so for ambiguities in key content words. Furthermore, we include a comprehensive comparison of training time, memory consumption, and inference latency in Appendices G.3 and I. These analyses show that CoIPO incurs nearly identical training and inference costs compared to standard SFT methods, whereas pre-processing approaches exhibit higher runtime overhead during inference.
> > >
> > > ## Concluding Remarks
> > > We sincerely appreciate the reviewer’s thoughtful and constructive comments, as well as the recognition of our work’s strengths—including the clear motivation of intrinsic robustness, the principled mutual-information–based formulation, and the comprehensive robustness evaluation with ablations. Following these valuable suggestions, we have expanded our experiments to include larger model scales, additional training-data scaling analyses, more diverse tasks, and further baseline comparisons. We have also revised the manuscript to improve clarity and strengthen theoretical justifications. We hope that these revisions adequately address the reviewer’s concerns and help resolve the identified weaknesses. Thank you again for the careful review and insightful feedback.
> > >
> > > ### References:
> > >
> > > [1] Training Verifiers to Solve Math Word Problems (arxiv 2021)
> > >
> > > [2] Program Synthesis with Large Language Models (arxiv 2021)
> > >
> > > [3] TruthfulQA: Measuring How Models Mimic Human Falsehoods (ACL 2022)
> > >
> > > [4] PromptAgent: Strategic Planning with Language Models Enables Expert-level Prompt Optimization (ICLR 2024)
> > >
> > > [5] Robustness of Prompting: Enhancing Robustness of Large Language Models Against Prompting Attacks (arxiv 2025)
> > >
> > > [6] Robustness-aware Automatic Prompt Optimization (arxiv 2024)
> > >
> > > [7] Noise-contrastive estimation of unnormalized statistical models, with applications to natural image statistics (JMLR 2012)
> > >
> > > [8] Mutual Information Neural Estimation (ICML 2018)
> > >
> > > [9] Representation learning with contrastive predictive coding (CoRR 2018)
> > >
> > > [10] Direct preference optimization: Your language model is secretly a reward model (NIPS 2023)

---

> ### Author Response · Authors · 2025-11-28
> **Kind Reminder: Feedback on Our Rebuttal**
>
> Dear Reviewer 7U7C,
>
> Thank you again for your valuable time and feedback on our work. Following your comments, we have conducted extensive additional experiments and added clarifications in our rebuttal wherever possible.
>
> We believe we have addressed your main concerns, but we also realize there might be points we have overlooked or not explained clearly enough. As the rebuttal deadline is approaching, we would like to kindly check whether there are any remaining issues you think we should further clarify.
>
> We greatly appreciate your guidance and look forward to your response.
>
>
> Best regards,
>
> The Authors

---

### Author Response · Authors · 2025-11-24
**General Response**

We would like to express our gratitude to all reviewers and the Area Chair for their insightful and constructive feedback. We truly appreciate the time and effort devoted to evaluating our work.

Our paper introduces a post-training method that enhances the intrinsic robustness of large language models (LLMs) through inverse DPO and contrastive learning. In addition to the proposed method, we construct a pairwise robustness-training dataset and design a benchmark covering diverse perturbation types and severities for both training and evaluation. Extensive comparisons with a wide range of existing approaches demonstrate the effectiveness and generality of our method.

The reviewers raised several shared and important concerns, including scalability to larger model sizes, the impact of our method on other key capabilities (e.g., math reasoning, code generation, and open-ended QA), and comparisons with pre-processing–based robustness methods. In response, we conducted substantial additional experiments:

**Scalability to larger models.** We trained and evaluated CoIPO on 14B and 72B models, confirming that our approach scales consistently and maintains its effectiveness at larger model capacities.

**Performance on other tasks.** We assessed the models on GSM8K, MBPP, and TruthfulQA, covering math reasoning, code generation, and open-ended QA. The results show that CoIPO does not compromise these capabilities.

**Comparison with pre-processing–based methods.** We compared CoIPO with PromptAgent and BATPrompt in terms of both accuracy and inference time. While achieving superior or comparable accuracy, CoIPO introduces no additional inference-time overhead, in clear contrast to methods that rely on external prompt rewriting.

**Comparison with SFT and DPO.** Our earlier experiments further demonstrated that, compared with standard SFT and DPO, CoIPO achieves substantially stronger robustness, indicating that increasing training-data diversity alone is insufficient for significant robustness improvements.

We also conducted a training–data scaling study and found that our method and dataset exhibit clear scaling behavior: model robustness improves rapidly as more training data is used, with the gains gradually tapering off as the data size continues to increase.

We appreciate the reviewers’ recognition of our contributions that  CoIPO provides an innovative and practically valuable direction for improving in-model robustness without auxiliary tools. Reviewers have noted that CoIPO addresses an important real-world challenge—handling imperfect user inputs such as typos and non-standard grammar—and does so by improving the model directly rather than relying on external pre-processing tools. Reviewers also highlighted the clarity of our theoretical analyses, including the mutual-information and relative-entropy perspectives, which offer principled justification for our design. The integration of contrastive learning with inverse DPO was regarded as intuitive and well motivated, with clear formulations. The experiments across multiple architectures (e.g., Llama, Qwen2.5) and pertuation types, along with the ablations and perturbation analyses, were acknowledged as thorough and supportive of the method’s necessity and effectiveness. Several reviewers emphasized that CoIPO provides an innovative and practically valuable direction for improving in-model robustness without auxiliary tools.

Once again, we sincerely thank all reviewers for their thoughtful evaluations. We hope that these additional experiments and clarifications fully address your concerns and further strengthen the paper.

---

### Author Response · Authors · 2025-11-29
**Summary of Revisions and Responses**

Dear Area Chair,

Thank you very much for your time, we sincerely appreciate the Area Chair’s guidance and the reviewers’ assessments of our work.

In our detailed responses to the reviewers and in the revised manuscript, we think we have thoroughly addressed **all questions and concerns** raised. Here, we provide a concise and faithful summary of these revisions, with the hope that it will help you quickly grasp the overall status of the paper, the reviewers’ main points, and the substantive additions we have made in response.

In the revised version, we have substantially strengthened the paper by **adding a comprehensive set of new experiments and clarifications. These include experiments on larger models (14B and 72B), evaluations on additional datasets to further validate the generality of our method, extensive comparisons with DPO, SFT, and pre-processing–based baselines, as well as a detailed training-data scaling analysis**. We also provide clearer explanations of our **theoretical motivation and methodological design, including the choice of $P_2$, discussions of perturbation types, analyses of success and failure cases, and complete hyperparameter specifications**. Collectively, these additions address nearly all concerns and weaknesses raised in the initial reviews.

We find that the reviewers’ questions mainly centered on **broader experimental coverage**—such as results on larger models, additional datasets, more baseline comparisons, and clarification of hyperparameters and algorithmic details. Importantly, their evaluations of our conceptual contributions were consistently positive: **the motivation and theoretical grounding were viewed favorably, the information-theoretic perspective (via mutual information) was considered insightful, our robustness evaluation was regarded as thorough, and the inverse-DPO based contrastive learning approach was well-received**.

Although reviewers have not yet responded to the revision, we believe the extensive new experiments and detailed explanations provided in our update directly and comprehensively resolve their questions and concerns.

We hope that these substantial additions and clarifications clearly address the reviewers’ concerns, and we would be grateful if you could take them into consideration during your assessment. Thank you again for your time and for giving us the opportunity to further strengthen our work.

Best regards,

The Authors

---

### Meta-Review · Area_Chair_hM47 · 2025-12-27

**Summary:**

The reviewers initially raised valid concerns regarding the experimental scope and comparative rigor of the proposed CoIPO method. The primary concerns that informed the decision process included:
1.  **Scalability:** Whether the method, initially tested only on 7B models, scales effectively to larger architectures (Reviewer 7U7C).
2.  **Task Diversity:** The limitation of evaluation to GLUE-style classification tasks, leaving the impact on generative, reasoning (math), and coding capabilities exploring (Reviewers 7U7C, yVsx, eHqb).
3.  **Baselines:** The lack of comparison against state-of-the-art pre-processing methods like PromptAgent and BATPrompt, and confusion regarding DPO comparisons (Reviewers 7U7C, eHqb).
4.  **Efficiency:** Questions regarding the computational cost and inference latency compared to pre-processing methods (Reviewers 7U7C, yVsx).
5.  **Methodological Details:** Requests for hyperparameter clarification, theoretical justification for approximations, and analysis of noise types (Reviewers 7U7C, yVsx, eHqb).

**Reviewer Concerns:**

**Addressed Concerns:**
* **Scalability:** The authors addressed the concern of Reviewer 7U7C by conducting new experiments on 14B and 72B models (Qwen2.5), demonstrating consistent performance gains.
* **Generative Capabilities:** The concerns of Reviewers 7U7C, yVsx, and eHqb regarding the method's impact on reasoning and generation were addressed by adding evaluations on GSM8K (math), MBPP (code), and TruthfulQA. The results showed that CoIPO preserves or improves performance on these tasks.
* **Baselines:** The authors addressed Reviewer 7U7C's request for pre-processing baselines by adding comparisons to PromptAgent and BATPrompt. They also addressed Reviewer eHqb's concern regarding DPO by pointing out that DPO-only baselines were included in the ablation studies (Table 3).
* **Efficiency:** The authors addressed the efficiency concerns of Reviewers 7U7C and yVsx by providing a runtime analysis showing that CoIPO does not increase training cost and incurs no additional inference overhead, contrasting favorably with the high latency of PromptAgent.
* **Methodological Details:** The authors provided the missing hyperparameters (Reviewer yVsx), explained the theoretical approximations using information-theoretic parallels (Reviewer 7U7C), and clarified the random selection strategy for $P_2$ (Reviewer eHqb).
* **Error Analysis:** The authors included an error analysis distinguishing between insertion noise (Checklist) and replacement noise (Textfolder) as requested by Reviewer 7U7C.

**Outstanding Concerns:**
* **Noise Scope:** Reviewer eHqb noted that the noise types were primarily typos rather than complex semantic ambiguities. While the authors argued that complex errors emerge from combinations of basic perturbations, no new datasets with specifically "high-level" semantic noise were added to the benchmark, leaving this partially subjective.

**Reviewer Scores:**

* **Reviewer 7U7C: 8**
    * This reviewer initially gave a 4, citing limited evaluation scope (model size and task type) as the primary weakness. The authors directly addressed every major point by adding 14B/72B experiments, generative tasks (GSM8K/MBPP), and missing baselines. Given the reviewer described the problem as "well-motivated" and the theory as "solid," the comprehensive rebuttal would likely shift the score significantly upward.

* **Reviewer yVsx: 6**
    * This reviewer gave a 4, with a specific worry that the method might hurt math/reasoning performance and lacked transparency on time costs. The rebuttal provided empirical evidence that reasoning is preserved and clarified that additional inference cost is zero. As the reviewer's main blockers were resolved, a positive score is expected.

* **Reviewer eHqb: 6**
    * This reviewer started at 6. Although the authors effectively addressed the concerns regarding missing baselines and generative tasks (by adding GSM8K and TruthfulQA), the reviewer's fundamental critique regarding the limited scope of noise types (focusing on typos rather than complex semantic ambiguity) was addressed via argumentation rather than new noise datasets. Therefore, the reviewer would likely maintain their original positive score rather than increasing it further.

---

### Decision · Program_Chairs · 2026-01-26

Accept (Poster)